# MEASURING AI "SLOP" IN TEXT

## ABSTRACT

AI "slop" is an increasingly popular term used to describe low-quality AI-generated text, but there is currently no agreed upon definition of this term nor a means to measure its occurrence. In this work, we develop a taxonomy of "slop" through interviews with experts in NLP, writing, and philosophy, and propose a set of interpretable dimensions for its assessment in text. Through span-level annotation, we find that binary "slop" judgments are (somewhat) subjective, but such determinations nonetheless correlate with latent dimensions such as coherence and relevance. Our framework can be used to examine AI-generated text in both detection and binary preference tasks, potentially offering new insights into the linguistic and stylistic factors that contribute to quality judgments. We highlight that fully automated and scalable methods remain an open challenge.

## 1 INTRODUCTION

"Slop" has emerged as a term describing generic, low-quality content that appears to have been generated by AI.[1] Recent news articles offer salient examples of such AI "slop", ranging from non-factual claims ("... add nontoxic glue to make cheese stick to a pizza", "geologists advise eating at least one rock a day"; Metz, 2024; Scott, 2024) to useless information ("fodder for websites whose only purpose appears to be optimising for [search engines]"; Mahdawi, 2025). Conversations on social media highlight indicators of "slop" in LLM responses, including overuse of certain terms, low information density, and structural quirks such as lists-as-responses.[2] Despite the sudden ubiquity of the term, there is no clear definition of, nor method, for *measuring* "slop" in text.

This gap matters: large-scale surveys, such as Microsoft's Occupational Implications of Generative AI (Tomlinson et al., 2025) and Anthropic's Economic Index (Handa et al., 2025) reveal AI is primarily used in writing and information gathering tasks. Defining and measuring "slop" may help characterize and ultimately improve LLM writing. Some individuals deeply familiar with AI generated content can reliably detect AI writing on the basis of structural and lexical quirks, even without training (Chakrabarty et al., 2024; Russell et al., 2025). Yet text can be perceived as "slop" even when not generated by AI, and not all AI-generated text reads as "slop".

Our primary aim in this work is to characterize qualities of texts that contribute to them being categorized as "slop." Such factors may explain instances where humans mistakenly characterize human-written text as AI-generated, and "slop" might provide an explainable metric that accounts for binary preferences between texts collected from human annotators. We apply principles from measurement theory to conceptualize and operationalize a definition of "slop" (Bandalos, 2018). We aim to provide language for articulating style and components of bothersome LLM-generated text, while also providing a framework for measuring such aspects.

Our main contributions are as follows: We first **introduce a working definition and taxonomy of "slop"** and map each dimension to automatic metrics where possible (§3). To validate this framework, we collect **span-level annotations from expert writers over 150 news articles and 100 question-answering passages** to provide a fine-grained analysis of slop indicators (§4). Although binary assessments of "slop" vary across individuals, we show that **our taxonomy helps explain which latent qualities (e.g., coherence, relevance, structural features) contribute most to these judgments** (§5). We also find that the strength of latent qualities vary based on domain, and that **our**

---

[1]Slop was on the shortlist of Oxford Dictionary's Word of the Year 2024, which claims a "332% increase" in usage: https://corp.oup.com/word-of-the-year/#shortlist-2024

[2]https://x.com/aidan_mclau/status/1884770586276381179

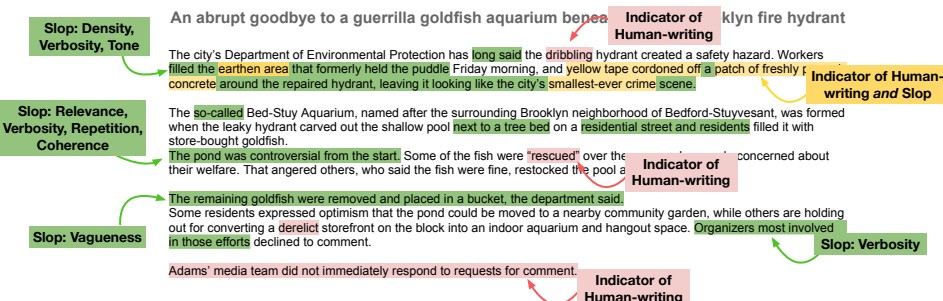

Figure 1: Sample of annotations over a *human-written* news article highlighting indicators of "slop" (red; from Russell et al. 2025), human-writing (green; ours), and both (yellow). "Slop" labels are notably different than indicators of human-written text.

**taxonomy provides a useful framework** for guiding quality assessment under different tasks (§5). For axes that lack adequate automatic measurements (e.g., relevance, coherence, fluency), we find that **standard text metrics fail to capture annotator preferences.** Finally, we show that capable reasoning **LLMs also fail to reliably extract and identify "slop" in text** (§6).

## 2   RELATED WORK

**AI-Text Detection.** There is now a small body of work on discriminating between human- and AI-written texts, e.g., DetectGPT (Mitchell et al., 2023) and Binoculars (Hans et al., 2024) provide scores for the likelihood that they were AI generated, and report high discriminant performance (0.95 AUROC). Russell et al. (2025) provide an interpretable guide listing key indicators of AI-written text. While related, recognizing "slop" differs from AI-text detection *in general*, and can be applied to any text source (whether AI-written or not). In this work our taxonomy and annotations diverge from those used for AI-text detection in general.

**Text Diversity.** Prior work has sought to characterize aspects of texts related to how repetitive and *templated* they are. Salkar et al. (2022) investigated repeated $n$-grams in LLM outputs in the context of summarization. Shaib et al. (2024b) found that modern LLMs are prone to repeatedly generate favoured *syntactic templates*, i.e., sequences of Part-of-Speech (PoS) tags. Padmakumar & He (2023) and Tevet & Berant (2020) examined lexical and semantic diversity in generated texts, introducing metrics to quantify variation across outputs and emphasizing its importance to generation quality. These existing efforts have informed the way in which we are thinking about what characterizes writing style and AI "slop" and provides automatic measurements for key aspects of "slop."

**Text Quality Measurements.** Text quality has typically been measured using simple surface-level metrics like BLEU (Papineni et al., 2002) and ROUGE (Lin, 2004), which can be effective when reference outputs are available (e.g., machine translation evaluation). More recent work has recognized that text quality is not monolithic but rather comprises multiple, sometimes competing dimensions that must be measured independently, and accordingly focused on multidimensional frameworks assessing properties of texts. Chakrabarty et al. (2025b) provide an editing taxonomy to correct (Chakrabarty et al., 2025a) recurring AI-writing flaws such as clichés and unnecessary exposition. Similarly, Bharadwaj et al. (2025) show that reward models over-weight 5 superficial writing cues including length, structure, jargon, sycophancy, and vagueness. Both works confirm that multiple factors contribute to text quality. Our work is complementary to measuring quality in general: We target stylistic patterns unique to LLM writing that are not covered by other taxonomies.

## 3   DEFINING LLM "SLOP"

The Oxford Dictionary defines "slop" as: *"[...] material produced using a large language model (LLM), which is often viewed as being **low-quality or inaccurate**. This type of low-quality, AI-generated material is becoming increasingly visible to people [...], who often view it as **unwanted or***

*inferior."* "Slop" as a construct does not immediately permit measurement: It is difficult to quantify "low-quality" or "unwanted" text. We propose a composite measure over observable characteristics of text, where we elicit salient characteristics from a set of individuals with a range of relevant expertise. Human writing can also read as "slop", but we adopt the above definition and focus on (seemingly) LLM-generated texts.[3]

We first solicited detailed definitions of "slop" from 19 individuals with a range of expertise across relevant disciplines including writing, journalism, linguistics, NLP, and philosophy (App. Table 4b). This group included PhD students, professors, and industry professionals from the listed disciplines. All but one respondent had 3 or more years experience in their field at the time of their response (App. Figure 4a). We asked individuals to describe their familiarity with the term "slop" in the context of AI-generated content, as well as a description of typical use (if any) of LLMs in their work. 11 experts (58%) had encountered the term "slop" as relates to AI-generated content. Most reported using LLMs more than 2 times a week ($n = 14$). The rest mostly used them sporadically ($n = 4$) with 1 expert never using them. We asked experts to provide a definition and list key characteristics of text that make it "slop." We provide the full survey sent to experts in Appendix A.

| Themes | Final Codes | Granular Codes | Count |
|---|---|---|---|
| **Info. Utility** | Density | IU1: Density | 5 |
| | Relevance | IU2: Relevance | 9 |
| **Info. Quality** | Factuality | IQ1: Factuality | 7 |
| | Bias | IQ2: Bias | 2 |
| **Style Quality** | Structure | SQ1: Repetition | 7 |
| | | SQ2: Templatedness | 2 |
| | Coherence | SQ3: Coherence | 6 |
| | (Aspects of) Tone | SQ4: Fluency | 4 |
| | | SQ5: Verbosity | 5 |
| | | SQ6: Word Complexity | 1 |
| | | SQ7: Tone | 3 |

Table 1: Themes and codes for slop, count of expert responses containing each.

Using qualitative content analysis and deductive coding techniques, we map expert definitions of "slop" to measurable concepts (Mayring, 2000). We begin by identifying key terms in survey responses and building a code list until saturation (i.e., until no new codes are created). We then map each response on to one of the following codes: Factuality, Information Density, Bias, Relevance, Repetition, Templatedness, Verbosity, Word Complexity, Tone, Coherence, Fluency, Diversity, Engagement, Vagueness, and Utility.

Assigned codes were separately reviewed by all authors, as were disagreements and redundant codes. The codebook was iteratively updated throughout this process. Redundant codes (e.g., Vagueness and Information Density) were collapsed. We further categorize codes with overarching categories or *themes*: Information Utility, Information Quality, and Style Quality. Table 1 describes the full code hierarchy within each theme, and the count of responses containing each code tag.

## 3.1 Datasets

We select two datasets to annotate for "slop", motivated by two practical observations: First, that LLM-written text is becoming commonplace in reporting news on the internet (Tomlinson et al., 2025; OpenAI, 2025). The second is motivated by the use-cases outlined by our experts in §3, where a majority ($n = 9$) reported using LLMs for question answering tasks (Appendix B).

**News Article Generation.** We evaluate "slop" over 150 news articles released by Russell et al. (2025), in which annotators are asked to label texts as being AI-written or not. Each unique article has a human-written source and a corresponding AI-written article, generated by either Claude (Anthropic, 2024), GPT-4o or o1-pro (Jaech et al., 2024). Additionally, each article includes a "humanized" article, where the above models are prompted to avoid obvious LLM-writing indicators.

**Retrieval-Augmented QA.** MS MARCO (Bajaj et al., 2016) is a large-scale machine-reading-comprehension benchmark comprising real Bing search queries. Each example contains an anonymized user query, a set of candidate web passages retrieved by the search engine, and a human-written answer. We randomly sample 100 queries and generate responses from Llama-4 Scout (AI,

---

[3]Interestingly, what is considered "slop" in human-written text can differ in characteristics, and may even serve as an intermediary in writing processes (cf. Appendix B)

2024), OLMo-2-13B Instruct (OLMo et al., 2025), Mistral-7B (Jiang et al., 2023), GPT-4o (OpenAI et al., 2024), and Gemma-2-27B (Team et al., 2024) (Generation details in Appendix C).

## 3.2 PILOT "SLOP" ANNOTATION

As an initial validation of the taxonomy, we hire 5 professional copy-editors from the Upwork platform[4] to annotate "slop" spans in two datasets: News article generation (Russell et al., 2025); and Retrieval Augmented QA (MS MARCO; Bajaj et al., 2016).[5] These datasets span different writing styles, expected passage lengths, and serve different purposes for the reader. We paid annotators at a rate of $35-45 USD an hour. Each article took ∼10-15 minutes to annotate, with an average of 871 words per article. The MS MARCO dataset took ∼4-7 minutes to annotate, with an average of 55 words per passage. We provided annotators with a guide containing codes for indicators of "slop" from our expert definition survey (§3; Appendix Table 7). We asked annotators to read the text in full, and first answer whether they initially perceive the text as "slop." We then had annotators label spans in the text (word-level) that instantiate any of the "slop" codes. Texts may have span annotations even if not initially deemed by the annotator to be "slop" overall. We provide the full set of questions given to annotators in Appendix Figure 6.[6] In sum, each annotator annotated the same 10 articles and 10 passages in the pilot round.

Annotation comprising multi-label, multi-span labelling is a difficult task and requires collaborative stages of task training and alignment among annotators. After independently completing the pilot round, annotators met to review codes and annotation strategy. Here, the guide was discussed in detail: clarifications around labelling strategy (e.g., whether to select only the most salient codes versus coding every feature), and terminology (e.g., Fluency as a measure of *language naturalness* versus correct grammar) were adjudicated. Most disagreements came from labelling strategy and terminology, rather than disagreements over the text spans. Table 1 shows the final themes and codes after annotator discussion.

## 3.3 FINALIZED "SLOP" TAXONOMY

Here, we describe each theme and code after annotator adjudication (See Appendix I for a description with examples).

**Information Utility** assesses how effectively a text conveys meaningful and contextually appropriate information. It comprises two key indicators: (i) **Density**, or the amount of substantive content relative to the length of the text, measured through information-theoretic token entropy (Meister et al., 2021) and propositional idea density (Brown et al., 2008), and (ii) **Relevance**, the alignment of content with task or prompt, measured through expert human annotations due to complexities in automated assessments (Clarke & Dietz, 2024).

**Information Quality** describes the accuracy and subjectivity of the presented information. **Factuality** assesses inaccuracies, hallucinations, or fallacious claims within the text, which require human annotations due to the complexity of automated factual evaluations in the absence of reference texts (Ramprasad & Wallace, 2024; Laskar et al., 2023). **Bias** (Subjectivity) assesses the presence or absence of a necessary subjective or rhetorical perspective, measured by the proportion of subjective words through an established lexicon (Wiebe et al., 2004).

**Style Quality** addresses properties related to expression and readability. Repetition, identified by lexical repetition metrics (Shaib et al., 2024a) and Templatedness, measured via syntactic structures (Shaib et al., 2024b) are key features of text **Structure**. **Coherence** is evaluated mostly via expert annotations due to the absence of reliable automatic measurements (Li et al., 2024; Murugadoss et al., 2024). Aspects of **Tone** evaluate the appropriateness and character of language relative to context, and include issues like excessive formality (Fanous et al., 2025; Yang et al., 2024). We include indicators such as Fluency (*naturalness* of language); Verbosity (passage and sentence length) (Zhang et al., 2024); and Word Complexity, i.e., use of unnecessarily complex vocabulary, measured by Gunning-Fog Index (Gunning, 1952) and Flesch-Kincaid Grade Level (Kincaid et al., 1975).

---

[4] https://www.upwork.com/

[5] Exempt Determination obtained from our institutional IRB. See Ethical Considerations for approval details.

[6] We built the labelling interface using a custom template in LabelStudio: https://labelstud.io/.

## 4 ANNOTATING FOR "SLOP" IN TEXT

**Annotation.** We select 3 annotators from our pilot study based on annotation quality and availability for the remainder of the datasets.[7] Each annotator reviewed 71 articles (total of 213 articles annotated), and 41 passages (total of 123 passages). We assign a subset of the same 45 news articles and 10 QA passages to all annotators for agreement assessment.

Measuring "slop" is difficult: Text can be assigned multiple labels, where a subset represent latent text features (Relevance, Bias, Coherence, Fluency, Tone), compared to directly measurable labels such as Verbosity and Repetition. Marchal et al. (2022) show that the expected overlap between any two annotators in a multi-label setting drops sharply as the number of labels and proportion of double-coded items increases, even after chance-correction. This agreement drops further still when labeling latent text features (e.g., coherence) that rely on annotators' mental models of these constructs. We follow prior work to select appropriate agreement measures for annotations.

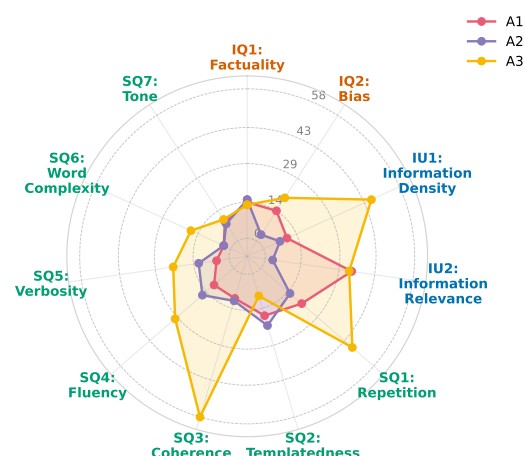

Table 2: Label counts and distributions for each annotator. Annotators used all codes, but there are individual differences in the code frequencies assigned in each theme.

**Span Agreement.** We use the span-level precision measure described in Chakrabarty et al. (2025b) to assess if annotators highlighted similar text. Span-level precision measures, for each annotator, the proportion of highlighted spans that overlap with another annotator's spans. Here, we consider sets of words for precision calculations At the paragraph level for articles annotators have a pairwise span-level precision of 0.80 (A1–A2), 0.65 (A1–A3), and 0.68 (A2–A3), indicating moderate to high agreement on problematic spans of text, regardless of the assigned code.

**Label Agreement.** We compute both Cohen's $\kappa$ and Gwet's $AC_1$ over the binary slop label, which indicates whether annotators agree on which documents are overall "slop." Annotator responses had a Cohen's $\kappa$ of -0.15 (A1–A2), 0.29 (A1–A3), and 0.06 (A2–A3), indicating poor to fair agreement. Reporting $\kappa$ is consistent with prior work in NLP, but we caution that these scores appear poor due to the low prevalence of the "slop" category.

| Themes | Final Codes | $\alpha_{MASI}$ | $\kappa$ | $AC_1$ | Prev. (%) |
|--------|-------------|-----------------|----------|--------|-----------|
| Info. Utility | Density | 0.34 | 0.16 | 0.45 | 59.1 |
| | Relevance | | 0.14 | 0.22 | 68.2 |
| Info. Quality | Factuality | 0.45 | 0.23 | 0.76 | 29.5 |
| | Bias | | 0.11 | 0.67 | 38.6 |
| Style Quality | Structure | | 0.11 | 0.51 | 52.3 |
| | Coherence | 0.34 | 0.13 | 0.39 | 59.1 |
| | Tone | | -0.11 | 0.20 | 50 |

Table 3: Agreement and label prevalence for "slop" codes.

Annotators assigned a positive label of "slop" to an average of 34% of the articles. By contrast, Gwet's $AC_1$ yields pairwise scores of 0.12 (A1–A2), 0.42 (A1–A3), and 0.28 (A2–A3), indicating fair to moderate agreement when correcting for prevalence. We ask annotators to assess "slop" labels *before* annotation, and posit that these overall assessments involve a degree of subjectivity. We do not necessarily expect strong agreement here.

**Taxonomy Agreement.** In the "slop" taxonomy labeling task, multiple codes can be assigned to a span, and multiple spans can exist in a document. We first aim to understand the convergence of the code sets assigned to each document. Following Marchal et al. (2022), we calculate Krippendorf's $\alpha_{MASI}$ which measures set agreement chance-corrected for partial overlaps. Next we try to evaluate the individual reliability of each code. We report both Cohen's $\kappa$ (for pairwise), Fleiss $\kappa$ (for three-

---

[7]Annotator background and details in Appendix E

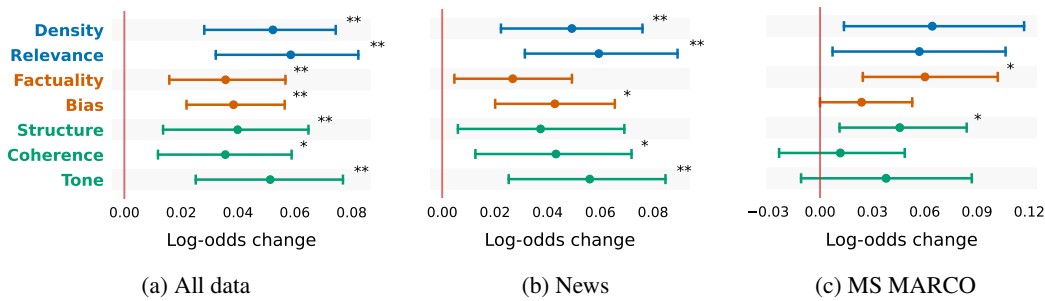

Figure 2: "Slop" codes most predictive of the overall positive label for (a) the entire corpus, (b) news, and (c) MS MARCO. * $p < 0.05$, and ** $p < 0.01$.

way) and Gwet's $AC_1$, noting that the $AC_1$ scores will be a more reliable assessment in this setting as annotators have differing rates of label assignment, shown in Figure 2.

Table 3 (all) and Appendix Table 12 (pairwise) report agreements calculated across finalized codes and overall themes. After three calibration rounds of guided discussion, we report theme-level Krippendorff's $\alpha_{MASI}$ of 0.34 (Info. Utility and Style Quality) and 0.45 (Info. Quality). These values fall within the "moderate-to-challenging" band ($\alpha \approx 0.10 - 0.50$) for high-entropy, construct-level annotation (Marchal et al., 2022), indicating that annotators consistently overlap on at least some taxonomy labels within each theme, but full label-set consensus is difficult to achieve. At the code level, Factuality ($AC_1 = 0.76$), Bias ($AC_1 = 0.67$), and Structure ($AC_1 = 0.51$) reach agreement above the 0.5 reliability threshold, indicating dependable annotation. In contrast, cognitively demanding constructs such as Coherence, Density, and Relevance fall closer to the "moderate-to-challenging" band, indicating that these labels can be used for research but with caution.

## 5 WHAT IS "SLOPPY" TEXT?

We now present results from all annotations collected across the news and QA datasets. Our analysis includes both a *combined* setting across all domains, as well as separate evaluations by domain (news vs. QA). For each setting, we report results aggregated across all annotators. (See Appendix I for individual plots). The combined analysis highlight common slop features shared across all data and annotators, while the disaggregated evaluations show variations that may arise from annotator subjectivity or domain-specific patterns.

We first construct "slop" features as an aggregated count and presence of the span-level codes across annotators. We fit a logistic regression model[8] with these features as the independent variables and the binarized slop label as the (single) dependent variable. This allows us to evaluate whether aggregated patterns in the span-level taxonomy are associated with the binary slop judgments. Features with adjusted $p < 0.05$ (after Bonferroni correction) are considered statistically significant predictors of whether annotators label texts as "slop." Figure 2 shows that the individual axes influencing slop assessments vary slightly in the domain-specific regressions.

### 5.1 RESULTS

We first confirm that more annotated spans in documents correlates with assessments of slop across annotators: $\rho_{news} = 0.70, \rho_{ms\_marco} = 0.51, \rho_{all} = 0.63$. Across the annotations, all seven codes are significant (positive) predictors of an item being labelled "slop," empirically validating the taxonomy built in §3. The strongest predictors are text issues like **Relevance** ($\hat{\beta} = 0.06$), **Density** ($\hat{\beta} = 0.05$), and **Tone** ($\hat{\beta} = 0.05$). The combined analysis of "slop" codes shows that broadly the quality deficit in the text is significant across all style, information quality, and utility themes (Fig. 2a). Text lacking relevance and information, or containing factual errors or biased language, is consistently labeled as "slop" across domains.

---

[8]Using `statsmodels`, version 0.14.0.

| Code category | Highlighted span |
|---|---|
| Relevance | "During the Roman Empire, physicians developed techniques to repair injured gladiators and soldiers, including methods for treating facial injuries and performing basic skin grafts. The field experienced a significant evolution during the Renaissance, as European surgeons began documenting and sharing their techniques more widely." |
| Factuality | "[...] leading to more frequent and severe heatwaves. "Climate change is like adding steroids to our weather," says Dr. Michael Oppenheimer, a climate scientist at Princeton." |
| Structure | "But did you know there's another important number-sort of like a "secret" code—printed just beneath the sell-by date? [...] Find the secret code, which is usually near the sell-by date." |
| Tone | "The very power of the word ["witch"] lies in its imprecision. It is not merely a word but an archetype, a cluster of powerful images." The uncertainty of exactly what a witch is forms part of the titillation" |

Table 4: Text marked by all annotators. Relevance: all marked as irrelevant. Factuality: the scientist exists but not attributed to this quote. Structure: marked for repeated text. Tone: marked for coherence, fluency.

**News.** For news articles, issues with style quality (Coherence, Tone), information utility (Density, Relevance), and Bias are significant predictors. Annotators deem text that is verbose, off-topic, or that contains tonal/framing issues in news articles as indicators of "slop" (Fig. 2b).

**MS MARCO.** For QA tasks, factuality and structural issues are the strongest predictors of "slop" for all annotators. Text from MS MARCO passages are short, which may reduce the significance of the Density, Relevance, and tonal codes. Answers that are concise, well-organized, and factually sound are valued more than polished prose (Fig. 2c).

Disaggregated analysis shows that features important for "slop" vary based on the purpose of the text: Factual and structural issues are significant for QA data, while stylistic and utility issues are prominent for news articles. **This distinction indicates the importance of evaluating LLM-written texts with respect to domain to contextualize their quality.** [9]

## 6 AUTOMATICALLY MEASURING SLOP

Building on the annotation analysis, we investigate whether assessments of "slop" can be measured with automatic methods. We evaluate standard text metrics and LLM-based approaches for capturing both the underlying dimensions reflected in the annotations and the overall "slop" assessments.

### 6.1 LINEAR MODELS

Table 6 provides the entire slop code taxonomy and a mapping to existing automatic text evaluation metrics. 3 out of 5 codes that were significant features of slop assessments in §5 do not have reliable metrics, motivating the need for human annotations. Nonetheless, we examine linear models with all available automatic metrics to assess their ability to capture the latent qualities of text in our taxonomy. Many automatic text measures have high overlap in information (e.g., Shaib et al., 2024a), shown in Figure 11, which can lead to multicollinearity issues in regression models.

To address this and handle class imbalance, we use $\ell 2$ regularization with $\alpha = 1.0$ and class weighting. We standardize all predictors and drop highly correlated features with threshold $\geq 0.95$.

**Results.** We measure the AUPRC curves for (a) News and (b) MS MARCO (App. Fig. 13) shows the AUPRC

| Dataset | AUPRC | Prevalence |
|---|---|---|
| News | 0.52 | 0.25 |
| MS MARCO | 0.55 | 0.27 |

Table 5: AUPRC across the News and MS MARCO datasets.

---

[9] We provide further qualitative assessment of the label distribution across topics and sources within the news domain in Appendix H.

| Themes | Final Codes | Codes | Sig. Feature? | Auto. Metric |
|--------|-------------|-------|:-------------:|--------------|
| Info. Utility | Density | IU1: Density | ✓ | Surprisal (Meister et al., 2021) |
|  | Relevance | IU2: Relevance | ✓ | — |
| Info. Quality | Factuality | IQ1: Factuality | ✗ | — |
|  | Bias | IQ2: Bias | ✓ | Subjectivity-Lexicon (Wiebe et al., 2004) |
| Style Quality | Structure | SQ1: Repetition | ✗ | Compression Ratios (Shaib et al., 2024a) |
|  |  | SQ2: Templatedness | ✗ | Templates-per-Token (Shaib et al., 2024b) |
|  | Coherence | SQ3: Coherence | ✓ | — |
|  | (Aspects of) Tone | SQ4: Fluency | ✗ | — |
|  |  | SQ5: Verbosity | ✗ | Num. Words |
|  |  | SQ6: Word Complexity | ✗ | GFI (Gunning, 1952) |
|  |  | SQ7: Tone | ✓ | — |

Table 6: Mapping of "slop" codes to existing automatic metrics. We mark the codes that are significant predictors for the slop label with a green checkmark: 3 out of 5 of the significant features do not have reliable automatic measures.

curves for (a) News and (b) MS MARCO. In both cases, the model has an AUPRC double the prevalence of the positive class, indicating that it captures some signal beyond random chance.

On News, the model achieves an AUPRC of 0.52 (prevalence is 0.25), while on MS MARCO it reaches 0.55 (prevalence is 0.27). The curves remain consistently above the prevalence baseline. These results suggest that the approach generalizes across the two domain, but that linear models are not sufficient for fully capturing the underlying signal.

## 6.2 Text Quality Reward Models

Given linear models and existing automatic metrics are not sufficient for fully capturing "slop" assessments, we now evaluate models trained elsewhere for writing quality. We use the Writing Quality Reward Model (WQRM; Chakrabarty et al. 2025a) to assign quality scores to our data. The model is trained on paragraph-level annotations, so we split all our News data into paragraphs.

**Results.** Appendix Figure 12 shows the distribution of WQRM scores over the News and MS MARCO datasets. Scores are distributed fairly broadly in both domains, ranging from around 5.0 to 8.5, with most documents clustering between 5.5 and 7.5, indicating medium to moderately-high quality writing (Chakrabarty et al., 2025a). Correlations with our slop annotations demonstrates that WQRM aligns with, but does not fully capture the "slop" label. WQRM shows lower correlation with the binary "slop" label: 0.25 for News, 0.15 for MS MARCO, suggesting it captures some signal of lower writing quality. When evaluating the number of annotated spans to the WQRM, correlation is 0.48 for News and 0.19 for MS MARCO, suggesting that as the number of issues in a text increases, writing quality decreases. These results indicate that WQRM captures some axes of "slop", especially in settings with multiple annotated issues.

## 6.3 Can LLMs Self Identify "Slop"?

Recent text evaluations have prompted LLMs to judge text qualities not readily captured by existing metrics (e.g., Liu et al. 2023; Zheng et al. 2023). This is usually done zero-shot, providing instructions for evaluation. Here we test three LLMs (GPT-5, Deepseek-V3, and o3-mini) for their ability to (a) predict binary "slop" labels and (b) extracting "slop" spans. In both cases, we provide the full "slop" guide given to annotators (See Appendix L for the full prompt given to each model).

**Results: Predicting Binary Slop Labels** Table 7a shows the results of binary slop prediction for GPT-5, Deepseek-V3, and o3-mini. Agreement with human annotators is low; $\kappa$ values are $\sim$0. Models under-predict the slop label (0.03-0.08), especially relative to humans (0.35). Both recall (0.08-0.12) and precision (0.25-0.42) are (very) low across all models, showing LLMs do a poor job at this task.

**Results: Extracting "Slop" Spans** On average, GPT-5 extracted longer text spans than human annotators (mean 58 vs. 41 characters, respectively). However, span-level alignment with human annotations is low. Table 7b shows the results of prompting GPT-5 zero-shot to extract spans, and

| Model | $\kappa$ | Pr./R | Pct. Slop |
|---|---|---|---|
| GPT-5 | 0.01 | 0.38/0.12 | 0.08 |
| Deepseek-V3 | -0.01 | 0.25/0.08 | 0.03 |
| o3-mini | 0.03 | 0.42/0.08 | 0.07 |
| Human | – | – | **0.34** |

(a)

| Model | k | Precision | Recall |
|---|---|---|---|
| GPT-5 | 0 | 0.14 | 0.11 |
| | 1 | 0.14 | 0.11 |
| | 3 | 0.16 | 0.13 |
| | 5 | 0.13 | 0.10 |
| Qwen-7B* | — | **0.32** | **0.30** |

(b)

Table 7: (a) Binary slop assessment: 0-shot prompting relative to human labels (humans assigned positive slop label to 35% of data). (b) Span-level extraction: Character-level precision and recall in zero- and few-shot settings. *Qwen-7B is fine-tuned.

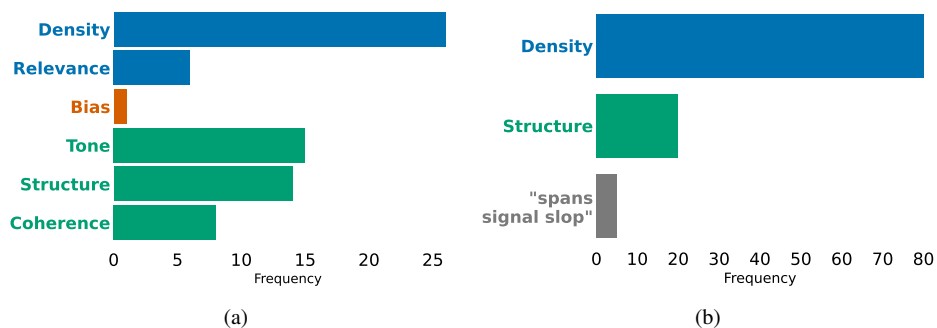

(a)                                   (b)

Figure 3: Frequency of (a) human-assigned slop code prevalence and, (b) "slop" category collapsed trigrams in o3-mini span rationales in the News domain. o3-mini over-represents issues related to Density in the rationales, while under-representing issues with Coherence, Tone, Bias, Relevance, and Factuality.

with in-context examples ($k \in [1, 3, 5]$). We report the *character-level* precision and recall.[10] GPT-5 achieves a precision of 0.08, recall of 0.12. Providing examples up to $k = 5$ does not improve precision nor recall by much: reporting precision of 0.13, and recall of 0.19. Further, the additional in-context examples do not significantly change which spans are extracted, there is a consistent F1 overlap of 0.65-0.67 between each $k$ setting and $k = 0$. While the higher recall relative to precision suggests the model can identify some relevant text spans, the overall overlap remains low.

### 6.3.1 LLM-Generated Span Rationales

In addition to extracting spans, the LLM-as-a-judge is instructed to provide a rationale for each selection. Reasoning chains do not reliably return the exact codes from the guide, so to assess the overall alignment of the reasoning and the human-assigned codes, we count and rank tri-grams from the judge-generated rationale (Figure 3, Appendix Figure 8).

Figures 3a and 3b highlight a mismatch between the human-assigned codes and those flagged most frequently by o3-mini. Specifically, o3-mini overwhelmingly focuses on Density-related issues, and does not show the full range of codes used by annotators. In Appendix G, we show that this does not significantly change when in-context examples are provided to the model.

### 6.4 Training Span Extraction Models

We trained a Qwen-7B reasoning model (DeepSeek-R1-Distill-Qwen-7B; DeepSeek-AI 2025) for slop span extraction. To provide rationales, we first generated silver annotations by prompting GPT-5 with the annotated span and label, asking for concise explanations of the label. We also augment our News and MS Marco data with data from LAMP Chakrabarty et al. (2025a;b), mapping their categories (such as cliche, redundant/exposition) into our slop taxonomy to create a consistent label

---

[10]Implementation details in Appendix K.

space. We filter the LAMP dataset to remove the creative writing subsets. We provide details of prompts used, label mappings, and training details in Appendix L.

**Results.** Evaluation on held-out data shows that the model achieves partial-overlap (character-level) scores of 0.33 precision, 0.22 recall, and 0.26 F1. Restricting to positive-only examples results in an F1 of 0.30 (precision 0.48, recall 0.22). The model also learns to abstain from predicting spans where there was no slop (similar to annotators), with an empty prediction rate of 44%. These results suggest that while the model can extract some slop spans with reasonable precision, it is still difficult to identify all issues. Training a model results in better performance than prompting GPT-5 with the guide, but neither perform especially well. This indicates a need for more research into metrics for identifying "slop" spans in texts.

## 7  DISCUSSION

LLMs are often deployed as cheap alternatives for human preference judgments in alignment and evaluation (Bharadwaj et al., 2025), however our findings highlight important limitations. Unlike reasoning tasks where rewards are verifiable, for subjective tasks there is a significant risk of mis-calibration. Prior work has documented these issues: Chakrabarty et al. (2025a) and Gooding et al. (2025), for example, show that LLMs struggle to select high-quality writing actions as judged by human experts, often treating suboptimal and optimal interventions as equally acceptable. This leads to low quality text that is often referred to as "slop".

A recent study from OpenAI Chatterji et al. (2025) shows that almost 50% of ChatGPT usage focuses on writing (28.1%) and information seeking (21.3 %) tasks. To ensure better alignment in such areas, we present the first systematic attempt at qualitatively characterizing "slop" in LLM-generated text. Our findings suggest that Information Quality, Information Utility, and Style Quality are important axes of text evaluation. Further, granular codes within each axis can vary in strength based on the domain, or the purpose of the text. We show that our taxonomy provides a useful framework for assessing writing across domains, beyond accuracy- or reference-based metrics. While overall "slop" judgments are somewhat subjective, our analysis shows that an increase in issues along these axes increases the likelihood of a text being judged as "slop". We also show that current evaluation practices are not sufficient for automatically measuring "slop". Existing automatic text metrics fail to capture whether generated text is genuinely useful or well-written relative to "slop". Neither LLMs-as-judges nor linear models trained over these features are able to fully approximate human assessments of "slop," however we hope the taxonomy can guide future improvements of LLM-based reward models. While our analysis focuses on text written in English, we hope the framework introduced here can support future analyses of other languages.

## 8  ETHICS STATEMENT

This study was reviewed and deemed exempt by our institutional IRB. Prior to their involvement in the project, all annotators were briefed on the purpose of the research and provided informed consent (Appendix D). We prioritized fair compensation which annotators set prior to participation. Our dataset consists of publicly available news and QA passages, and no personally identifiable information was used. The focus of this work is on characterizing properties of AI-generated text; it does not target or analyze individuals but rather professional assessments of writing quality.

## 9  LIMITATIONS AND REPRODUCIBILITY STATEMENT

The taxonomy, annotation guidelines, survey protocols, and detailed descriptions of data sources are included in Appendix A, F, and L. Code for data processing and "slop" assessments from LLMs are provided in the supplementary materials. All experiments are described with hyperparameters and settings in §6 and Appendix J. We additionally plan to release anonymized annotation data (with calculated automatic measures as described in §6.1) under an open-source license to facilitate replication of our results. We collect annotations on only English texts and acknowledge this is an inherent limitation of the work. While we expect the hallmarks of "slop" to be consistent across languages, future work should empirically verify this. We believe our work provides data and a framework to facilitate comparisons of "slop" evaluations in other languages

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

# A  DEFINITION COLLECTION

We provide the full survey sent to annotators in Table 9. Here, we show the full set of questions and answer options in the survey. Annotators explicitly provided permission at the end to share anonymized and aggregate responses to the survey.

| Question | Answer Options |
|---|---|
| What is your primary field of work? | Natural Language Processing
Linguistics
Writing (e.g., Copywriting, Journalism)
Other: [Free text] |
| How many years of experience do you have in your field? | 1-2
3-5
5-8
9+ |
| Have you encountered the term "slop" before, as it relates to generated text or images? | Yes
No |
| If you answered "Yes" to the above question, please describe which contexts you've encountered the term in (e.g., in a news article, in a podcast, in discussions) | Free text |
| How often do you use large language models (LLMs) in your work? | Never
Rarely (sporadically)
Sometimes (2 times a week)
Often (3-4+ times a week) |
| If you use LLMs once or more a week, please select the type of tasks you use them for. | Ideating
Writing
Rewriting
Summarizing documents/texts
Creative writing
Question Answering (general)
Question Answering (specific, based on an input document)
Other: [Free text]
*(Check all that apply)* |
| Please define "slop" as it relates to either AI-generated or human-written text. Please be as specific as possible. Think of the contexts in which you use or encounter LLM-generated texts to help guide your answer. | Free text |
| Which key characteristics in text are associated with "slop"? | Free text |
| Is all AI-generated text "slop"? | Yes
No
Maybe |
| Any other thoughts you would like to share about the definition of "slop" or its role in text? | Free text |

Table 9: Anonymous survey sent to experts to collect definitions of "slop."

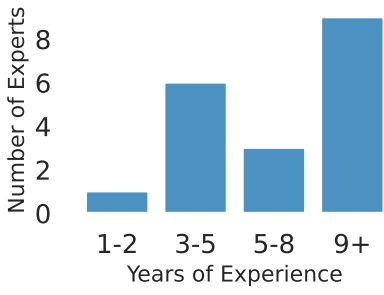

| Field of Work | Count |
|---|---|
| Natural Language Processing | 6 |
| Writing | 5 |
| Engineering | 2 |
| Machine Learning | 3 |
| Linguistics | 2 |
| Philosophy | 1 |
| **Total** | 19 |

(a) Years of experience for each expert.

(b) Work areas of experts surveyed.

Figure 4: Expert demographics: (a) distribution of years of experience and (b) fields of expertise.

| **Expert Response (trunc)** |
|---|
| "...'slop' text from humans as something that is **very generic and overly verbose**—perhaps **excessive marketing copy** or **rambling prose** that is published when it really should have been heavily edited. I wouldn't call it 'slop' if it was simply a first draft—it's the **brazen publishing** (online or IRL) of content that **wastes a reader's time** that befits the term best." |
| "The classic **highschool transitional words** that students just learning to write essays use, **without variance**." |
| "Slop in certain human contexts may perform a **useful function as an intermediary step** in the author's process (consider certain kinds of **notes or student writing**, for example)." |
| "...there are many responses from humans that are **hastily written without critical thoughts**." |

Table 10: Sample responses describing how "slop" may manifest in human written text.

## B    ADDITIONAL SURVEY FINDINGS

Experts provided their years of experience in their reported fields (Table 4b), which we report in aggregate in Table 4a. Most annotators had $\geq 3$ years of experience, and many had professional experience in the field of NLP or writing. Additionally, experts identified characteristics of "slop" that can appear in human-written text, but all point to qualities that serve a different purpose than those identified for AI-generated text.

## C    TEXT GENERATION DETAILS

Here we describe the data generation procedure for the News and MS MARCO datasets. For News, we use the articles first introduced in Russell et al. 2025: these are news articles generated by GPT-4o, Claude-3.5- Sonnet, and O1-pro. The articles are generated by providing the title of a real news article pulled 8 American publications: Associated Press, Discover Magazine, National Geographic, New York Times, Reader's Digest, Scientific American, Smithsonian Magazine, and Wall Street Journal. [11] For MS MARCO, we randomly sample a subset of 100 passages.[12] We filter the passages for answers longer than 30 words to ensure long enough responses for annotation. We prompt 4 models (OLMo-2-7B-Instruct, Mistral-7B, Gemma-27B, GPT4o-mini) to generate a response using the following prompt:

```
You are given a search query and a set of potentially
relevant articles.  Your task is to answer the query.
Sources:  [SOURCES] Query:  [QUERY]
```

---

[11] https://github.com/jenna-russell/human_detectors
[12] https://huggingface.co/datasets/microsoft/ms_marco/

Where we replace `SOURCES` and `QUERY` with the relevant context and query from the dataset. For all models (where possible), we greedily generate the responses (e.g., setting the parameter for sampling to `False`). For open-source models, we use the HuggingFace platform to generate the text.[13]

## D  CONSENT FORMS

All annotators were briefed on the study and provided explicit consent to share their anonymized responses. We provide the consent form given to participants in Figure 5.

## E  EXPERT ANNOTATOR DETAILS

Here, we provide details of the backgrounds and expertise of our selected annotators. Our analyses are limited to English texts so we hired annotators who are fluent in English. All annotators are native English speakers from North America. Each annotator had between 15 and 30 years of experience working as professional copy-editors and writers, in Education, Publication, and Business fields. Annotators have worked additionally as writers for print and online professional publications, and as educators teaching writing.

## F  ANNOTATION DETAILS

We build our annotation interface using LabelStudio, shown in Figure 6. Annotators are instructed to first answer the "Initial Assessment" of "slop." After this question is answered, they proceed with the annotation with the codes. We provide annotators with a PDF document containing the specific definition of slop (Fig. 7. This guide also contains a definition of each "slop" code with examples of how each code can appear in text (Table 11).

Table 11: "Slop" Codes with Examples

| Code | Name | Description | Example |
|------|------|-------------|---------|
| **Information Quality** | | | |
| IQ1 | Factuality | Incorrect or fabricated information Misleading or fallacious claims | "Dr. Sarah Johnson of Harvard University published groundbreaking research on this topic in 2022." *(Slop if Dr. Johnson doesn't exist, isn't at Harvard, or didn't publish such research)* |
| IQ2 | Bias | Text that feels too "objective" when subjectivity is appropriate Missing rhetorical point of view when needed Lack of appropriate perspective based on context Content that seems detached when engagement is required The presence of inappropriate perspective or assumptions | "The economic policy changes of 2023 were universally beneficial." *(Slop because it presents a one-sided view of complex policy impacts)* |
| **Information Utility** | | | |

*Continued on next page*

---

[13]https://huggingface.co/

Table 11 *(continued)*

| Code | Name | Description | Example |
|------|------|-------------|---------|
| IU1 | Information Density | Text that is verbose but conveys little actual information
Generic statements that could apply in almost any context
Excessive filler words and phrases that add no value | "In today's fast-paced modern world of cutting-edge technology and innovation, it has become increasingly important to consider the various factors and elements that contribute to our understanding of this complex and multifaceted issue."
*(Slop because it uses many words to say almost nothing)* |
| IU2 | Information Relevance | Content that fails to address the nuances of the query or task
Content that contributes nothing meaningful to context/query/task
Text that appears disconnected from its intended purpose
For text with additional context, consider relevance to such texts
For text with no additional context, consider internal relevance | In response to "How can I improve my marathon time?":
"Running is an excellent form of exercise with many health benefits including improved cardiovascular function, enhanced mood, and weight management."
*(Slop because it doesn't address the specific question about improving marathon times)* |
| **Style Quality** | | | |
| SQ1 | Repetition | Excessive use of the same words or phrases
Redundant statements that add no new information
Overuse of transitional phrases common in formulaic writing
Low diversity in vocabulary and expression | "The project was a success. The team accomplished their goals successfully. The successful outcome was due to the team's hard work."
*(Slop due to repetition of "success/successful" without adding new information)* |
| SQ2 | Templatedness | Over-reliance on formulaic structures and patterns
Predictable formatting patterns (e.g., excessive use of bullet points)
Standard transitional phrases used repeatedly
Frequent appearance of text following a common pattern | "Dr. Smith, a researcher at Oxford University, found that... Professor Johnson, a scientist at Cambridge University, discovered that... Dr. Williams, an expert at Yale University, confirmed that..."
*(Slop because it follows the same formula repeatedly)* |
| SQ3 | Coherence | Poor sentence structure or organization
Inconsistencies in argument or narrative
Text that requires significant effort to follow
How paragraphs work together to advance the argument or story | "Climate change is affecting global temperatures. Polar bears are mammals. Ice cream melts in warm weather. Arctic ice is melting. Some people enjoy winter sports."
*(Slop because the sentences, while related to temperature, don't flow logically)* |

Table 11 *(continued)*

| Code | Name | Description | Example |
|------|------|-------------|---------|
| SQ4 | Language Naturalness | Language that sounds artificial or manufactured
Strange turns of phrases or unnatural language
Technically correct grammar that still reads unnaturally
Misaligned word choice for the context
Can co-occur with verbosity if long, does not necessarily include complex words | "The earthen area that formerly held the puddle was now dry."
*(Slop because natural language would simply say "The puddle had dried up" or "The ground where the puddle had been was now dry")* |
| SQ5 | Verbosity | Excessive wordiness relative to the information conveyed
Unnecessarily "flowery" or descriptive language
Text that prioritizes word count over meaningful content
Long-winded explanations that need significant editing | "The consumption of the aforementioned beverage, which had been prepared with the utmost care and attention to detail by the skilled barista, provided me with a sense of satisfaction and contentment that permeated my entire being."
*(Slop because it could simply say "I enjoyed the coffee")* |
| SQ6 | Word Complexity | Inappropriate use of vocabulary relative to context
Unnecessary jargon or complicated terminology
Content filled with buzzwords that obscure meaning
Overuse of rare words | In a general article about gardening: "The phenolic compounds in certain cultivars exhibit antimicrobial properties that mitigate pathogenic microorganism colonization."
*(Slop because it uses unnecessarily complex terminology for the intended audience)* |
| SQ7 | Tone | Generic voice lacking character or purpose
Missing perspective or point of view
Overly formal language in casual contexts (or vice versa)
Text that reads like an outside observer rather than engaged writer
Overconfidence in response
Can have a relationship with factuality (IQ1) | In a blog post about personal travel experiences: "The aforementioned destination offers numerous recreational activities for tourists. Visitors may engage in swimming, hiking, or dining at local establishments."
*(Slop because it uses an inappropriately formal tone for a personal blog)* |

Careful selection of measurements to operationalize the definition of "slop" requires consideration of construct validity (i.e., whether we are measuring the intended phenomena), and a discussion of any errors the measurement may inadvertently introduce. Here, we describe the constructs comprising "slop" and provide an example of how to operationalize each; we rely on prior work, where possible, for established methods of measuring each code introduced. Note that we aim to establish the validity of a combination of such measures to quantify "slop", rather than focusing on whether individual measures alone capture their intended construct.[14]

## INFORMATION UTILITY

**Density.**    A key indicator of "slop" is the relatively low density of information within it. Such texts are often verbose without conveying much information, or contain many generic statements

---

[14]Hence our reliance, where possible, on prior works on quantifying the individual factors considered.

## Slop Evaluation Consent and Information Form

**B**  *I*  U̲  ⸢  ⸝

▮▮▮▮▮▮▮▮▮▮▮▮▮▮▮▮▮▮▮▮▮

**Name of Investigator(s):** ▨▨▨▨▨▨▨▨▨▨

**Title of Project: Survey on Identifying "Slop" In Text**

**Request to Participate in Research**

We would like to invite you to participate in a web-based online survey. Our goal is to annotate spans of text that constitutes slop, and the preference for the appearance of these spans. "Slop", as it refers to AI-generated content, is a vaguely defined term synonymous with "low-quality, inauthentic, or inaccurate." (see: https://corp.oup.com/word-of-the-year/#shortlist-2024).

Your task is to review short texts from various domains and tasks (e.g., news articles, biomedical abstracts, creative writing) and determine whether they constitute "slop" in your opinion.

For each piece of text, you will be asked to (1) read the text, (2) highlight any instances of the text that you deem "slop" according to the definition guide we provide you, and (3) explain why the highlighted sections indicate presence or absence of "slop."

We will first ask you to complete a short questionnaire that identifies your familiarity with AI-generated text. At the end of all the tasks, we will also ask you to complete a short reflection on your impressions while annotating.

We are asking you to participate in this study because you have indicated that your primary language is English. **You must be at least 18 years old to take this survey.**

**The decision to participate in this research project is voluntary.** You do not have to participate and you can refuse to answer any question. Even if you begin the web-based online survey, you can stop at any time. **The possible risks or discomforts of the study are minimal. There are no direct benefits to you from participating in this study.** However, your responses may help us learn more about measuring aspects of style in written text.

**You will be compensated at a rate of $20-50 per hour.**

**Your part in this study is anonymous to the researcher(s). However, because of the nature of web-based surveys, it is possible that respondents could be identified by the IP address or other electronic record associated with the response. Neither the researcher nor anyone involved with this survey will be capturing those data. Any reports or publications based on this research will use only group data and will not identify you or any individual as being affiliated with this project.**

**Future Use of Data or Biospecimens:** Your de-identified information could be used for future research **without additional informed consent.**

**If you have any questions regarding electronic privacy,** please contact ▨▨▨▨▨▨▨▨▨▨▨▨▨▨▨▨

**If you have any questions about this study,** ▨▨▨▨▨▨▨▨▨▨▨▨▨▨▨▨▨▨▨▨

**If you have any questions regarding your rights as a research participant,** please contact ▨▨▨▨▨▨▨▨▨▨▨▨▨

**This study has been reviewed and approved by the** ▨▨▨▨▨ **Institutional Review Board.**

**By submitting this form below you are indicating that you consent to participate in this study. Please print out a copy of this consent screen or download a copy of the consent form for your records.**

Thank you for your time.

▮▮▮▮▮▮▮▮

Figure 5: "Slop" Evaluation consent form given to annotators prior to the study.

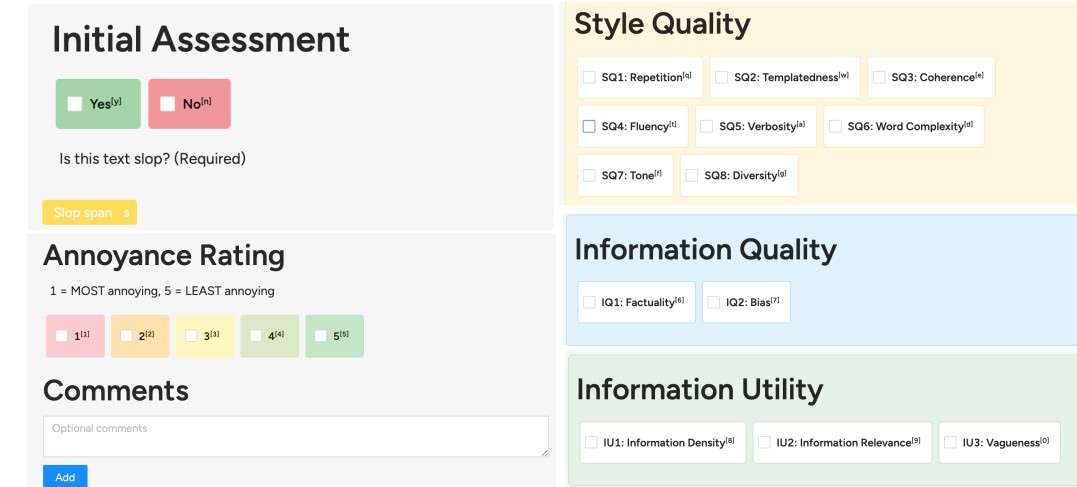

Figure 6: questions given to annotators in interface (LabelStudio)

that are broadly applicable. We measure information density in two ways. In the first, we adopt an information-theoretic approach, following Meister et al. (2021). The uniform density hypothesis posits that speakers generally tend towards spreading information uniformly across utterances. We measure token entropy using GPT-2 (Radford et al., 2019), and then evaluate the mean and coefficient of variation. A higher mean indicates an overall lower information density in text, and a higher coefficient of variation indicates less uniformity, both of which can be indicative of "slop".

In the second measurement, we measure the propositional idea density. Ideas can be approximated by the number of verbs, adjectives, adverbs, prepositions, and conjunctions, and the density can be estimated by adjusting the counts of sets of high-likelihood part-of-speech sequences and dividing by the total number of words in a document Brown et al. (2008). Higher values of idea density indicate a higher amount of information in the text.

**Relevance.** Measuring relevance (to a context), similar to factuality, is an active research area. Most methods assume access to a high-quality set of source documents and queries. Relevance, where context is provided, is measured relative to the query and task at hand. In the absence of additional context (e.g., task or domain), relevance can be evaluated on the internal consistency of the passage. "Slop" can comprise content that fails to address the query or task, sometimes subtly. Recently, Clarke & Dietz (2024) showed that GPT-4o cannot reliably act as a replacement for human assessments of relevance for conditional generation. Therefore, we rely on human assessments of relevance with additional context provided where possible.

## INFORMATION QUALITY

**Factuality.** In non-fiction texts, high-quality text is accurate. LLM "slop", however, is defined as having "subtle inaccuracies", introducing hallucinations ("non-existing entities"), or containing fallacious claims. Automatically measuring factuality is an open research problem (e.g., Ramprasad & Wallace 2024; Laskar et al. 2023), and can depend on whether reference (source) documents are available. We rely on human annotations to detect inaccuracies in LLM-written texts in all cases.

**Bias (Subjectivity).** Bias in text can refer to a range of topics that might influence the subjectivity of writing, and can span social (Blodgett et al., 2020) or cognitive (Atreides & Kelley, 2024) facets. Unless explicitly prompted to produce an opinion, much of the content in "slop" lacks subjectivity in presenting information (factual or otherwise). Of the expert definitions that mention bias, there is a notable focus on the *lack* of subjectivity in "slop." There is often a missing rhetorical point of view when it is otherwise needed, or a lack of appropriate, engaged perspective. For instance, an LLM-generated movie review that simply states facts such as "[...] movie received 3.5 stars and had a small budget" does not provide any of the subjective assessments one expects in a critique.

# Definition

> "Slop" refers to AI-generated text that is low-quality. It can appear superficially correct but is some combination of generic, overly verbose, inaccurate, irrelevant to its intended purpose, and contributing little meaningful value to the reader (despite sounding fluent). "Slop" typically displays patterns of repetition, formulaic structure, vague language, and an absence of authentic perspective. Not all AI-generated text is "sloppy," and human writing can be sloppy too.

Below, we outline all the key text characteristics (as codes) that contribute to "slop". Please refer to this guide when labeling spans of text for "slop." A span may fall under multiple codes.

## Annotation Instructions

1. Read each text in full before making any annotations.
2. Make an initial assessment: Is this text "slop"? (yes/no)
3. If yes, highlight specific spans of text and assign appropriate code(s).
4. Use the annoyance scale (1-5) where 1 = most annoying (requires complete rewriting) and 5 = least annoying (minor issue).

Please note that recognizing text as AI-written doesn't automatically make it sloppy! You may find "slop" in parts of a text while still answering "no" to the overall "slop" assessment if these issues are minor.

## General Guidance

- Focus on quality issues, not just AI detection.
- Consider the context, purpose, and intended audience of the text when making judgments.
- Code for the most significant issues rather than tagging everything possible.
- When in doubt about which code to use, choose the one that best captures the core problem.
- For ambiguous cases, briefly note your reasoning.

Figure 7: Definition Guide presented to annotators (along with Table 11) to reference.

We use the subjectivity lexicon from Wiebe et al. (2004), which provides words with labels as either subjective (weak, strong) or objective. Following prior work, we define our bias measurement as the proportion of subjective words to total number of words in a document.

STYLE QUALITY

**Repetition.** When defined in the context of "slop", repetition entails excessive use of the same words or phrases and low diversity in vocabulary and expression. Prior work has looked at measuring

semantic (Tevet & Berant, 2020; Namuduri et al., 2025) and lexical repetition (Shaib et al., 2024a). We focus specifically on lexical repetition metrics, measuring the compression ratio over words (CR) and over parts-of-speech (CR: PoS) to capture repetitive phrases and words.

**Templatedness.** LLMs tend to write formulaically at the syntactic level (Shaib et al., 2024b). "Slop" may include an over-reliance on formulaic structures and patterns, such as predictable formatting (e.g., bullet points) and repeated use of certain transitional phrases. Following Shaib et al. (2024b), we measure the template rate and templates-per-token for text.

**Coherence and Fluency.** Automatically measuring coherence and fluency in text is difficult (Li et al., 2024; Murugadoss et al., 2024), and may require human assessments to validate. Fluency is the correctness of the written language. Coherence refers to the logical flow and connection between ideas presented in a text. State-of-the-art LLMs that have undergone rounds of post-training and alignment rarely produce text that is completely disfluent.[15] "Slop", however, can exhibit low coherence (such as poor sentence organization, inconsistencies in argument or narrative, or written in a way that demands significant effort to follow), or subtle disfluencies (e.g., strange turns of phrase, technically correct but unnatural language, or word choices misaligned to the context). We rely on expert human annotations to identify instances of disfluency or incoherence in texts.

**Verbosity.** LLMs tend to respond to simple queries with high verbosity, leading to training with explicit instructions to "be concise!" (Zhang et al., 2024). In "slop", texts are often highly verbose. We measure verbosity as the passage length (number of words), and also as the average length of sentences.

**Word Complexity.** Word complexity assesses the vocabulary in a passage relative to the context: "Slop" can contain unnecessary jargon, buzzword-laden content, or can exhibit an overuse of "rare" words (Hovy, 2016). Our evaluation of "slop" is in English texts, so we opt to use established measurements of complexity: Gunning-Fog Index, which measures the years of formal education needed to understand text on a first reading (Gunning, 1952), Flesch-Kincaid Grade Level (Kincaid et al., 1975), measuring the (U.S.) school grade level one needs to understand the text, Flesch Reading Ease (Flesch, 1948), measuring textual difficulty on a 100-point scale where higher scores indicate easier-to-read text. We also measure sentence and word length (Dale & Chall, 1948), as these directly correlate with text complexity.

**Tone.** The overall tone of a text should reflect an appropriate style and voice given the context. "Slop" may be read as lacking character or perspective, and as containing overly formal language in casual contexts. This can sometimes appear as overconfidence in responses, or sycophancy (Fanous et al., 2025; Yang et al., 2024). We rely on human annotators to identify a combination of this characterization of tone in "slop".

## G  HUMAN-LLM SPAN OVERLAP (QUALITATIVE ASSESSMENT

Figure 8 shows the top ranked trigrams, and their categorization along the "slop" themes by colour. o3-mini tends to overly reason about information density relative to human-assigned labels.

## H  RESULTS BY DOMAIN AND TOPIC

In the News domain, we further assess the distribution of "slop" labels stratified by the source of the article (e.g., Discover, Wall Street Journal). Figure 9 shows the distribution of labels from annotators across each News source. We find that the categories are roughly represented similarly across sources (e.g., Style Quality codes annotated at a much higher rate relative to Information Quality within each source). We include the source counts in Table 13.

---

[15]At least in English; Multilingual assessments may show otherwise.

| Themes | Final Codes | Pair | AC$_1$ | $\kappa$ | Prev. (%) |
|--------|-------------|------|--------|----------|-----------|
| Info. Utility | Density | A1–A2 | 0.92 | 0.37 | 9.1 |
| | | A1–A3 | 0.09 | 0.03 | 56.8 |
| | | A2–A3 | 0.14 | 0.08 | 54.5 |
| | Relevance | A1–A2 | 0.46 | 0.16 | 40.9 |
| | | A1–A3 | 0.18 | 0.21 | 68.2 |
| | | A2–A3 | 0.06 | 0.06 | 59.1 |
| Info. Quality | Factuality | A1–A2 | 0.88 | 0.61 | 18.2 |
| | | A1–A3 | 0.70 | 0.04 | 25.0 |
| | | A2–A3 | 0.70 | 0.04 | 25.0 |
| | Bias | A1–A2 | 0.81 | 0.19 | 18.2 |
| | | A1–A3 | 0.49 | 0.00 | 38.6 |
| | | A2–A3 | 0.70 | 0.13 | 25.0 |
| Style Quality | Structure | A1–A2 | 0.67 | 0.02 | 27.3 |
| | | A1–A3 | -0.43 | -0.05 | 79.5 |
| | | A2–A3 | -0.22 | 0.07 | 77.3 |
| | Coherence | A1–A2 | 0.83 | 0.33 | 18.2 |
| | | A1–A3 | 0.17 | 0.08 | 54.5 |
| | | A2–A3 | 0.06 | -0.01 | 59.1 |
| | Tone | A1–A2 | 0.70 | 0.04 | 25.0 |
| | | A1–A3 | 0.77 | 0.10 | 20.5 |
| | | A2–A3 | 0.76 | 0.23 | 22.7 |

Table 12: Gwet's AC$_1$, Cohen's $\kappa$, and prevalence for each annotator pair and final code.

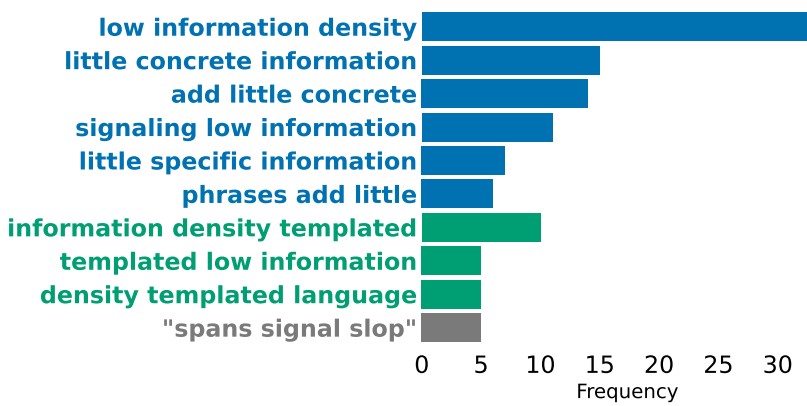

Figure 8: Tri-grams extracted from o3-mini rationales over highlighted "slop" spans.

## I   RESULTS BY INDIVIDUAL

We provide the pairwise agreement among annotators for all the "slop" codes in Table 12, including the percentage of overall prevalence of the label. A1/A2 had consistently strong agreement, whereas A2/A3 diverged. In adjudication meetings, annotators discussed these differences which can be attributed to editing style.

For all data, we find that the three annotators varied in which "slop" codes most strongly predicted their overall judgments (Fig. 10). For A1, information-related issues were more salient: Density, Relevance, Factuality, and Bias all showed strong positive associations. This suggests that A1 relied heavily on signs of low information quality or utility when identifying slop. A2, by contrast, was

| Publication | Number |
|---|---|
| National Geographic | 41 |
| Smithsonian Magazine | 38 |
| New York Times | 37 |
| Wall Street Journal | 31 |
| Discover | 29 |
| Readers Digest | 24 |
| Associated Press | 22 |
| Scientific American | 19 |
| Reader's Digest | 7 |

Table 13: Number of articles for each News source.

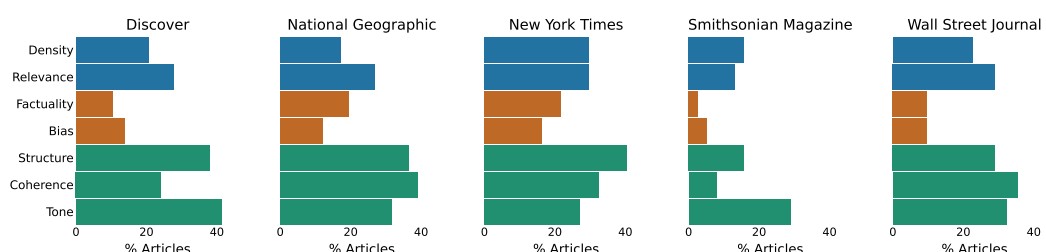

Figure 9: "Slop" axis label prevalence stratified by news source.

more selective, with verbosity (Density) emerging as the only significant predictor and Structure and Coherence showing positive though non-significant effects, indicating greater emphasis on how text was organized rather than on factual accuracy or bias. For A3, none of the codes reached significance, and while Density, Relevance, and Bias trended positive, wide confidence intervals suggest less consistency in how the taxonomy was applied. Taken together, these results highlight that annotators converge on verbosity as a core indicator of slop but diverge in how strongly they weight other dimensions such as Factuality, Bias, and Coherence.

## J  AUTOMATED METRICS

We report the correlation between automatic text metrics in Figure 11. Many metrics have moderate to high correlations indicating shared information.

We also report the distribution of WQRM scores in Figure 12 split by the (a) News and (b) MS MARCO datasets. The distribution of scores in the News domain is relatively broad. By contrast, the MS MARCO dataset shows a narrower spread, with most scores clustering between 5.5 and 7.0 and fewer documents at the extremes.

We compute and show the AUPRC for the automatic metrics using scikit-learn[16] in Figure 13. We train logistic regression models with $\ell2$ regularization using the liblinear solver. Features are standardized with a StandardScaler, and highly correlated features are removed with a threshold of 0.95. Models are tuned over a grid of $C \in \{0.01, 0.1, 1, 10\}$. We balance class weights.

## K  CHARACTER-LEVEL PRECISION AND RECALL

We provide the pseudo-code for calculating character-level precision and recall for span overlap in Algorithm 1. We note that this can be modified to calculate word-level overlap, and empirically find our conclusions hold when using both character- and word-level evaluations.

---

[16]https://scikit-learn.org/stable/

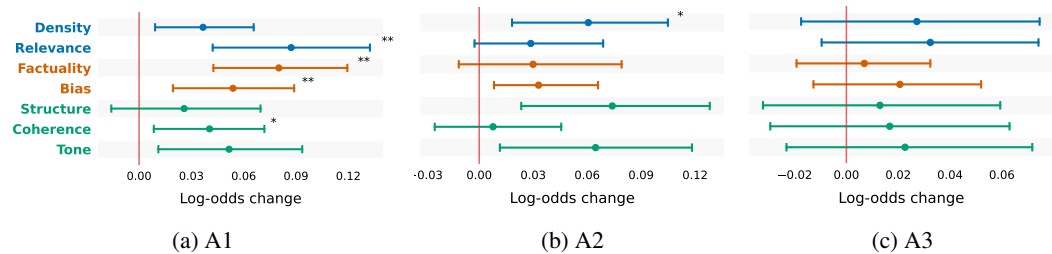

(a) A1        (b) A2        (c) A3

Figure 10: "Slop" codes most predictive of the overall positive label for (a) Annotator 1, (b) Annotator 2, and (c) Annotator 3 in the News domain. * $p < 0.05$, and ** $p < 0.01$.

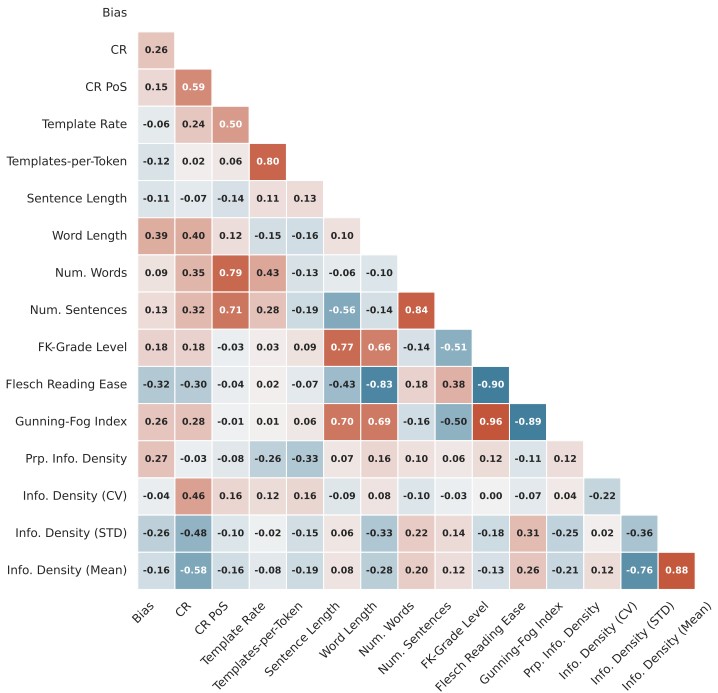

Figure 11: Correlation news

---

**Algorithm 1** Span-level Precision, Recall, and F1 Computation

1: gold_masks ← []
2: pred_masks ← []
3: **for** each row in $df$ **do**
4:      $text \leftarrow row.$text
5:      $gold\_spans \leftarrow$ parse_spans($row.$gold)
6:      $gold\_mask \leftarrow$ mark_characters($|text|, gold\_spans$)
7:      append $gold\_mask$ to gold_masks
8:      $pred\_spans \leftarrow$ parse_spans($row.$pred)
9:      $pred\_mask \leftarrow$ mark_characters($|text|, pred\_spans$)
10:      append $pred\_mask$ to pred_masks
11: **end for**
12: $g \leftarrow$ concatenate(gold_masks)
13: $p \leftarrow$ concatenate(pred_masks)
14: $tp \leftarrow$ count($g = 1 \land p = 1$)
15: $fp \leftarrow$ count($g = 0 \land p = 1$)
16: $fn \leftarrow$ count($g = 1 \land p = 0$)
17: $precision \leftarrow \begin{cases} \frac{tp}{tp+fp} & \text{if } tp + fp > 0 \\ 0 & \text{otherwise} \end{cases}$
18: $recall \leftarrow \begin{cases} \frac{tp}{tp+fn} & \text{if } tp + fn > 0 \\ 0 & \text{otherwise} \end{cases}$
19: $f1 \leftarrow \frac{2 \cdot precision \cdot recall}{precision + recall}$
20: **return** all computed metrics

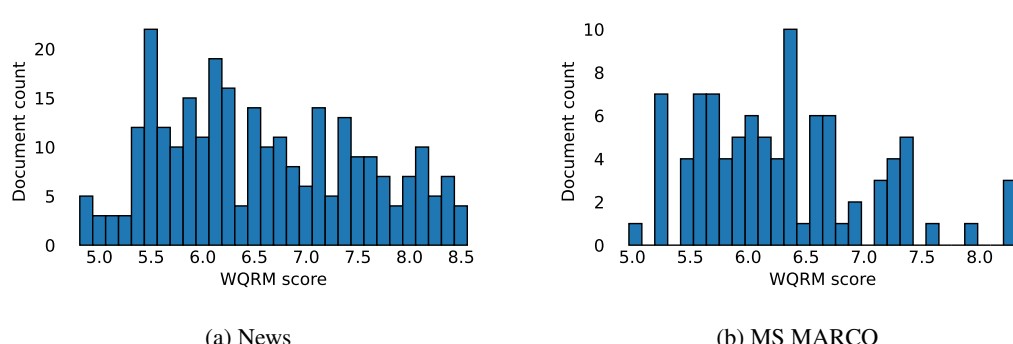

(a) News  (b) MS MARCO

Figure 12: Distributions of WQRM scores across the (a) News and (b) MS MARCO datasets.

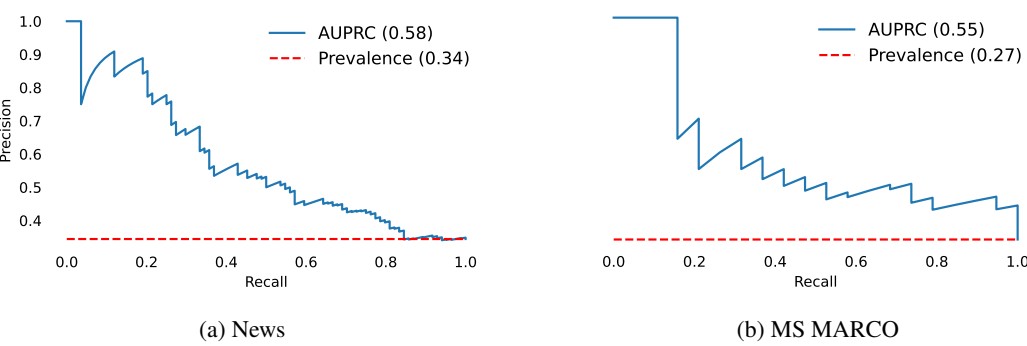

(a) News  (b) MS MARCO

Figure 13: AUPRC for linear models of all available automatic text metrics (Table 6). Prediction is almost double the prevalence rate in both datasets, but not sufficient as a standalone predictor.

## L LLM EVALUATIONS

### L.1 TRAINING

We trained Qwen-7B-reasoning for 5 epochs. We used a learning rate of $2 \times 10^{-4}$ with bf16 precision. To address class imbalance, we applied a positive oversampling rate of 0.5. We used the following prompt to guide answers during training:

```
You are a careful copy editor.  Given a paragraph,
extract the minimal set of short verbatim spans
(quoted) that are indicative of slop according to the
guide, then provide a brief reasoning.  The guide is
provided below.  Slop refers to AI-generated text that
is low-quality.  It can appear superficially correct
but is some combination of generic, overly verbose,
inaccurate, irrelevant to its intended purpose, and
contributing little meaningful value to the reader
(despite sounding fluent).  Slop typically displays
patterns of repetition, formulaic structure, vague
language, and an absence of authentic perspective.

[[Truncated]]

Return a JSON ONLY, no prose, with keys exactly as
follows:
{spans:  ..., reasoning:  ...}
```

where the **[[Truncated]]** section has a copy pasted version of the codes and their examples/definitions (Fig. 11).

| LAMP Category | Slop Taxonomy |
|---|---|
| Cliche | Tone |
| Poor Sentence Structure | Coherence |
| Awkward Word Choice and Phrasing | Tone |
| Tense Inconsistency | Tone |
| Unnecessary/Redundant Exposition | Density, Repetition |
| Lack of Specificity and Detail | Relevance |

Table 14: Mapping of the categories in Chakrabarty et al. 2025a to the "slop" taxonomy.

To extract the silver-label rationales, we use the following prompt on o4-mini reasoning models to gather reasoning chains:

```
SYSTEM_PROMPT = """
You are an experienced copy-editor.
For each numbered span you receive, write **one
sentence (≤ 25 words)**
explaining why the span is low-quality "slop," using
its FINAL CODE as the label.
Return the rationales in exactly the same numbered
order|nothing else.
""".strip()
SLOP_GUIDE = """
"Slop" = AI-generated text that is generic, verbose,
inaccurate, irrelevant, or
adds little real value.  It often shows repetition,
formulaic structure, vague
language, and no authentic perspective.
FINAL CODES (7-way collapse)
• Density { Many words, little information; filler or
fluff.
• Relevance { Off-topic or tangential to the
passage/question.
• Factuality { Incorrect, fabricated, or misleading
statement.
• Bias { One-sided, over-general, or unnuanced claim.
• Structure { Repetitive or templated sentence /
formula pattern.
• Coherence { Disjointed or ill-logical flow; hard to
follow.
• Tone { Awkward fluency, needless jargon, verbosity,
or style unsuited
  to context/audience.
""".strip()

TASK = """ Give numbered rationale(s) ( 25 words)
per span.  First, state the span label, then the
rationale.
Output **only** the rationale list|no extra commentary
as a python LIST. """.strip()
```

## L.2   DATA AUGMENTATION (LAMP)

For the LAMP data Chakrabarty et al. (2025a), we first filter for text in either the Travel Writing, Food Writing, or Creative Non-Fiction categories to match our News and QA data settings. We then map the labels map the following categories to our "slop" taxonomy (Table 14).

## L.3 PROMPTING

For prompting off-the-shelf GPT and DeepSeek models in zero- and few-shot settings, we used the following prompt(s).

```
SYSTEM_PROMPT_SPANS = (
  "You are a careful copy editor.  Given a paragraph,
extract the minimal set of short "
  "verbatim spans (quoted) that are indicative of \"slop\"
according to the guide, then provide a brief reasoning.\n"
  "The guide is provided below.  "Slop" refers to
AI-generated text that is low-quality.  It can appear
superficially correct but is some combination of generic,
overly verbose, inaccurate, irrelevant to its intended
purpose, and contributing little meaningful value to the
reader (despite sounding fluent).  "Slop" typically displays
patterns of repetition, formulaic structure, vague language,
and an absence of authentic perspective."
  "Factuality:  Incorrect or fabricated information,
Misleading or fallacious claims.  Example:  "Dr.  Sarah
Johnson of Harvard University published groundbreaking
research on this topic in 2022." (Slop if Dr.  Johnson
doesn't exist, isn't at Harvard, or didn't publish such
research)"
  "Bias:  Lack of appropriate perspective or
over-standardization.  Example:  "The economic policy
changes of 2023 were universally beneficial." (Slop because
it presents a one-sided view of complex policy impacts)"
  "Information Density:  Text that is verbose but
conveys little actual information.  Excessive filler
words.  Example:  "In today's fast-paced modern world
of cutting-edge technology and innovation, it has become
increasingly important to consider the various factors
and elements that contribute to our understanding of this
complex and multifaceted issue." (Slop because it uses many
words to say almost nothing)"
  "Information Relevance:  Appropriateness to the specific
context, query, or task.  For text with no additional
context (e.g., an article), consider internal relevance
within the passage.  Example:  In response to "How can
I improve my marathon time?":  "Running is an excellent
form of exercise with many health benefits including
improved cardiovascular function, enhanced mood, and weight
management." (Slop because it doesn't address the specific
question about improving marathon times)"
  "Repetition:  Excessive use of the same words or phrases.
Low diversity in vocabulary and expression.  Example:  "The
project was a success.  The team accomplished their goals
successfully.  The successful outcome was due to the team's
hard work." (Slop due to repetition of "success/successful"
without adding new information)"
  "Templatedness:  Over-reliance on formulaic structures and
patterns.  Predictable formatting patterns (e.g., excessive
use of bullet points).  Frequent appearance of text that
follows a common pattern (e.g., "Mr.  X, a Y-year-old Z").
Example:  "Dr.  Smith, a researcher at Oxford University,
found that... Professor Johnson, a scientist at Cambridge
University, discovered that... Dr.  Williams, an expert at
Yale University, confirmed that..." (Slop because it follows
the same formula repeatedly)"
  "Coherence:  Poor sentence structure or organization.
Text that requires significant effort to follow.  Example:
"Climate change is affecting global temperatures.  Polar
bears are mammals.  Ice cream melts in warm weather.  Arctic
ice is melting.  Some people enjoy winter sports." (Slop
```

```
      because the sentences, while related to temperature, don't
      flow logically)"
        "Fluency:  Strange turns of phrases or unnatural language.
      Example:  "The earthen area that formerly held the puddle
      was now dry." (Slop because natural language would simply
      say "The puddle had dried up" or "The ground where the
      puddle had been was now dry")"
        "Word Complexity:  Unnecessary jargon or complicated
      terminology.  Overuse of rare words.  Example:  In a general
      article about gardening:  "The phenolic compounds in certain
      cultivars exhibit antimicrobial properties that mitigate
      pathogenic microorganism colonization." (Slop because it
      uses unnecessarily complex terminology for the intended
      audience)"
        "Tone:  Appropriate voice and style for the context.
      Example:  In a blog post about personal travel experiences:
      "The aforementioned destination offers numerous recreational
      activities for tourists.  Visitors may engage in swimming,
      hiking, or dining at local establishments." (Slop because it
      uses an inappropriately formal tone for a personal blog)"
        'Return a JSON object:  { "spans":  ["...","..."],
      "reasoning":  "..." }'
      )
      SYSTEM_PROMPT_LABEL = (
        "You are a careful copy editor.  Given a piece of text,
      return a binary assessment of whether this is overall slop.
      "
        "The guide is provided below.  "Slop" refers to
      AI-generated text that is low-quality.  It can appear
      superficially correct but is some combination of generic,
      overly verbose, inaccurate, irrelevant to its intended
      purpose, and contributing little meaningful value to the
      reader (despite sounding fluent).  "Slop" typically displays
      patterns of repetition, formulaic structure, vague language,
      and an absence of authentic perspective."

        [[...  slop codes and examples ...]]

      Task:  Is this slop (0 = no, 1 = yes)
```

Where **[[...  slop codes and examples ...]]** has the full slop guide (definitions and examples) formatted into the text.