# OpenReview forum: "Measuring AI "Slop" In Text"
_ICLR.cc/2026/Conference — Submitted to ICLR 2026_

### Official Review · Reviewer_XZAE · 2025-10-30

**Soundness:** 2
**Presentation:** 2
**Contribution:** 2
**Rating:** 4
**Confidence:** 3

**Summary:**

In this paper, the authors aim to define and measure what they call AI slop for AI-generated text. They first design a taxonomy of slop from expert editors and operationalize it into several annotation dimensions covering information utility, information quality, and style-related issues. They then apply this taxonomy to news and QA dataset, and collect span-level and document-level annotations from 3 annotators to study how often these slop phenomena occur and how consistently humans can identify them. Finally, they explore whether slop can be detected or scored automatically, and report that current automatic methods only barely reproduce human judgments.

**Strengths:**

1. The paper focuses on an important and increasingly visible issue. Social media and the internet are increasingly flooded with AI-generated text, making it crucial to develop accurate methods for measuring and detecting low-quality AI content.

2. Even though it’s subjective, the three dimensions (Information Utility, Information Quality, Style Quality) give a structured way to talk about  why a piece of text feels like “AI slop” rather than relying on vague and binary judgments.

3. The paper is generally well written and easy to follow.

**Weaknesses:**

1. The core empirical claim (that the proposed taxonomy explains human judgments of “slop”) relies on annotations from only three final annotators. With such a small pool, rater-specific preferences and biases cannot be averaged out, and the already low cohen's kappa values (down to −0.15) become very difficult to interpret. The paper cites Marchal et al. to argue that “multi-label is hard,” but that is precisely the scenario where more annotators are needed to stabilize labels and reduce idiosyncratic noise. In fact, with such a limited rater pool, even adding a single additional annotator (e.g., moving from 3 to 4) could significantly shift agreement scores, especially considering the annotation task is very subjective.

2. If I understand correctly, the slop label is produced by the same annotators who provide the span level slop codes. As a result, the regression simply uses annotators’ own fine-grained judgments to predict their own overall slop decisions. This setup is somewhat confusing and circular, because it tests whether annotators agree with themselves rather than validating the taxonomy against an independent signal. Of course the p-values for predictors  should be singifiantly as it just check the internal consistency between annotators' span-level subjective codes and their binary slop decisions. Moreover, even under this weird design, the agreement scores remain low, which further raises concerns about the stability and reliability of the proposed construct.

3. While conceptually interesting, the proposed taxonomy does not presently yield a practical or scalable method for detecting AI slop. All attempted automatic approaches, including traditional machine learning models, logistic regression, and LLMs-as-judges, either barely outperform the base rate or fail to meaningfully agree with human judgments at all (LLM cohen's kappa essentially zero).  For a venue like ICLR, this shows a substantial gap between the stated goal of a measurable framework and the actionable evidence of model signal.

**Questions:**

see above weakness

---

> ### Author Response · Authors · 2025-11-21
>
> We thank the reviewer for their thoughtful feedback, and appreciate that they agree our framework provides a structured method for assessing text quality judgements! We address the weaknesses raised by the reviewer below. We have also added changes to the manuscript (written in pink) based on the feedback.
>
> **W1:** We clarify that our annotators are trained domain experts, and we prioritized multiple training and adjudication sessions relying on their expertise **(now included in Appendix E)**. We also would like to highlight that we report multiple agreement metrics that are perhaps more robust to few-rater designs, and carefully discuss the various agreement scores on each part of the annotation (e.g., spans, binary labels, and various granularities of the codes; L241, 247, 264) and find in our experiments that the overall pattern of label distributions remains consistent. We agree that future work should increase annotations, and we release our initial data and code to enable future studies at scale to validate and improve our taxonomy with larger and different reader populations (non-experts).
>
> **W2:** To clarify, we do not predict an annotator’s binary label from the span codes. Instead, we infer document-level attributes by aggregating span occurrences and counts, and then use these attributes to predict binary “slop” labels. The regression tests whether the inferred span attributed map to the overall “slop” judgements generalized across annotators, not whether the annotators predict their own marks. We thank the reviewer for this point, and have added this clarification in the methods section **(Section 5, L305-311)**.  While the agreement is modest, we believe this is informative. It indicates that the span locations are consistent but the mapping to the global judgement depends on subjective thresholds. This indicates that slop should be treated as perhaps a continuous score rather than a binary assessment, which is why we present both assessments in this work.
>
> **W3:** We believe presenting these results are an important finding of this work. LLMs are increasingly being used to judge vague notions of text quality (e.g., [1] directly shows the high disagreement between LLM-judges and humans), and we caution that, even when given human-constructed and validated guidelines of different qualities of text, LLMs fall short of these types of assessments. Additionally, we highlight that existing automatic metrics that are not LLM based are not sufficient for capturing the more complex (and not directly measurable) construct of “slop”. We also believe the PR(AUC)s presented in our analyses indicate a mixed, but non-trivial signal for evaluation improvement.
>
> Our findings here provide grounding and motivation for future work to develop reliable metrics to measure such qualities of text. We thank the reviewer for noting this gap, and agree that this should be reflected in our stated goals. We have updated the abstract and introduction **(L017-019; L069)** to better represent our findings.
>
> [1] Bavaresco, A., Bernardi, R., Bertolazzi, L., Elliott, D., Fernández, R., Gatt, A., ... & Testoni, A. (2025, July). Llms instead of human judges? a large scale empirical study across 20 nlp evaluation tasks. In Proceedings of the 63rd Annual Meeting of the Association for Computational Linguistics (Volume 2: Short Papers) (pp. 238-255).

---

> > ### Author Response · Authors · 2025-11-28
> >
> > We would like to follow up on our rebuttal in case any further clarification would be helpful. We appreciate your time and engagement!

---

### Official Review · Reviewer_8fEQ · 2025-10-31

**Soundness:** 3
**Presentation:** 3
**Contribution:** 2
**Rating:** 4
**Confidence:** 3

**Summary:**

The paper defines and measures “AI slop”, generic, low-quality AI text, by proposing a taxonomy spanning information utility (density, relevance), information quality (factuality, bias), and style quality (structure, coherence, fluency/verbosity/tone). Using expert-guided, span-level annotations on news and QA corpora, the authors show that all seven factors predict human “slop” judgments, but their importance shifts by domain (e.g., relevance/density in news; factuality/structure in QA). Inter-annotator agreement is modest overall yet strongest for factuality, bias, and structure. Automated proxies (repetition, compression, readability) and a reward model provide only limited signal, and current LLM-as-judge approaches perform poorly at both binary slop detection and span extraction.

**Strengths:**

- The paper proposes a three-axis taxonomy (Information Utility, Information Quality, Style Quality) and argues that increases in these issues raise the odds a text is judged “slop”.  It establishes a foundational framework for future work by standardizing terminology and guiding annotation and metric design.
- Logistic regressions show code salience shifts by domain: utility/style and bias matter more in news; factuality/structure dominate in QA.
- Linear models over standard metrics achieve only modest AUPRC; several salient codes lack reliable automatic measures. LLM-as-judge underperforms.

**Weaknesses:**

- Linear models over available metrics achieve only modest AUPRC (0.52–0.55), underscoring that current proxies don’t capture “slop” robustly.
- Annotations are confined to two English domains (news and MS MARCO QA), leaving external validity for other genres (e.g., creative, technical) and languages unclear.
- Final labeling relies on three professional copy-editors whose preferences differ; authors note divergences attributable to editing style.

**Questions:**

- How exactly should annotators (or practitioners) decide the overall yes/no “slop” label before doing spans.
- For practitioners, do you recommend a decision rule that combines the seven codes into a yes/no label (e.g., weighted sum, top-k issues), or is the binary label strictly a human judgment for now?

---

> ### Author Response · Authors · 2025-11-21
>
> We thank the reviewer for their feedback and for recognizing the value of our framework! We discuss the weaknesses and questions below.
>
> **W1:** We view this not as a limitation but rather an important finding from our analysis: There is a gap between the notion of low quality text and our existing metrics for measuring textual properties along various dimensions. We offer a range of experiments with more recent methodologies (e.g., trained LLMs, LLMs-as-judges) to bolster this point. We emphasize that not all results need to be “positive”; it is important, in our view, to understand limitations of current methods as well.
>
> **W2:** We agree that the feature strength may change under different domains, and in different languages. However our hope is that this provides a taxonomy for measuring various aspects of text and their impact on “slop” assessment.
>
> **W3:** We conduct multiple training sessions with all annotators to ensure alignment on the editing style; however even with the degree of subjectivity, we find that annotators agree on problematic spans in the text, motivating our analysis of the codes.
>
> **Q1:** Our study captures implicit human judgments of slop, revealing how subjective notions of text quality guide the overall judgements. We collected annotations for the binary “slop” label in this manner to assess judgements unbiased by the guide and annotations. We note that this allows us to use the correlation between the number of “slop” spans and binary judgments to validate our taxonomy. Practically, practitioners may prefer to use this framework to compute continuous scores (rather than binary assessments) of slop (see our response to the next question).
>
> **Q2:** The binary slop label reflects a direct human judgment rather than a deterministic aggregation of the seven span-level codes. Each code captures a distinct dimension of low-quality text and our focus here was to characterize how these dimensions relate to overall human impressions of “slop”. Future work should assess this question more directly by collecting larger-scale judgments along each axis and modeling how combinations of issues predict global “slop” judgments. Our labeled data can help with developing and validating mappings between span-level indicators and holistic quality judgments.
>
> For practitioners, we would actually encourage considering the continuous score assigned to a piece of text, which provides more information than the binary assessment of “slop” or not, though this would depend on exactly how the user wanted to use the measurement.

---

> > ### Author Response · Authors · 2025-11-28
> >
> > We would like to kindly follow up on our rebuttal. Please let us know if we can provide any further clarification. Thank you for your time!

---

### Official Review · Reviewer_zmra · 2025-10-31

**Soundness:** 3
**Presentation:** 4
**Contribution:** 3
**Rating:** 6
**Confidence:** 4

**Summary:**

This paper focuses on establishing a definition and measurement for "AI slop", which is a term for low-quality AI-generated text. The authors first develop a taxonomy of "slop" by interviewing 19 experts from fields including NLP, writing, and philosophy. This taxonomy organizes "slop" into roughly three main themes: Information Utility, Information Quality and Style Quality. The authors then validate this framework by collecting span-level "slop" annotations from expert writers on two datasets: news articles and question-answering passages.

The paper finds that binary "slop" judgments are somewhat subjective and these judgments strongly correlate with the latent dimensions in the taxonomy. The strongest predictors of a text being labeled "slop" are issues with Relevance, Density, and Tone. The authors also find that these predictors vary by domain: news articles are penalized for style and utility issues, while QA passages are penalized more for Factuality and Structure. Finally, the authors find that automatic detection of "slop" is difficult and LLMs perform poorly at both binary "slop" classification and span extraction.

**Strengths:**

- The paper is well-written and focuses on a timely problem, the lack of a formal definition and measurement for low-quality AI-generated content.
- Directly interviewing experts and codifying their responses to develop the "slop" taxonomy is an appropriate choice here, as they are the most reliable informants of what slop actually is.
- The finding that "slop" characteristics are domain-dependent is interesting, and adds nuance to the measurement of slop. The paper shows that predictors for "slop" in news articles differ from those in QA.
- The finding that current LLMs are unreliable judges of "slop" has important implications as it shows a limitation of using LLMs for subjective text evaluation.

**Weaknesses:**

- The paper argues that measuring “slop” can “help characterize and improve LLM writing” but doesn't actually show empirical evidence for this. It could have been interesting to show proof that slop measurements can help us train models that are less "sloppy".

- There is very limited discussion of how the measurement of slop interacts with prior work in NLP that measures properties of text such as diversity, novelty, complexity etc. For instance, the authors could have used existing automatic metrics for these properties (like the ones listed in Table 6) to measure slop.

- There is limited evaluation of the validity of the proposed taxonomy. The codification done based on interviews seems comprehensive but is not further validated and can hence seem a bit ad-hoc at times. Is the taxonomy sufficient? If not, how can it be improved?

- “We select 3 annotators from our pilot study based on annotation quality and availability for the remainder of the datasets.” - This statement is a bit vague about the background and expertise of these 3 annotators. Since the majority of the annotation was done by these 3 annotators, I think providing more context about them would be helpful. Finally, just relying on 3 annotators weakens the conclusions of the paper significantly.

- I think the main paper should describe precisely how span-level agreement between annotators is measured since it is an important part of the main methodology. Span boundaries are bound to have low agreement between annotators, so how does the paper deal with this? Please also describe it in your author response.

- The proposed taxonomy relies heavily on codes like Relevance, Coherence, and Factuality, which the authors mention lack reliable automatic measures. The fine-tuned Qwen-7B model for span extraction also shows weak performance. This makes the framework difficult to use for large-scale automatic evaluation.

- The paper shows that binary "slop" judgments are subjective, which is supported by the low inter-annotator agreement on the binary label. While this is a finding, it also makes it difficult to create a scalable measure for a "slop" measurement.

**Questions:**

- The agreement for the binary "slop" label was low but the span-level location agreement was moderate to high. This suggest that annotators agree where problems exist but disagree on whether those problems are sufficient to label the whole document "slop". Could you discuss this? Does this suggest "slop" is better measured as a count of span-level issues rather than a binary document label?
- The annotation guide says that "Not all AI-generated text is 'sloppy' and human writing can be sloppy too". Did the annotators label the human-written source texts from the news dataset? If so, were human-written "slop" spans identified, and did their characteristics differ from the AI-generated "slop" spans?
- Why do you think LLMs like GPT-5 perform poorly on this task? These models are optimized for human preferences but they seem unable to detect these "slop" characteristics. Is it a blind spot in current RLHF practices or a more fundamental limitation of LLM-as-judges for subjective style evaluation?

---

> ### Author Response · Authors · 2025-11-21
>
> We thank the reviewer for their feedback and are glad they recognized the value in the problem we chose and the validity of our coding methods. We address both the weaknesses and questions raised by the reviewer below. We have also updated the manuscript to address suggestions, written in pink.
>
> **W1:** We agree that using measures of “slop” to reduce its production is a key practical aim. However, to do this we first need to be able to automatically measure the degree to which a generated text reads as “slop”: This is the focus of our work here. If we could reliably recognize “slop”, it seems clear that one could then use, e.g., DPO to train models to discourage production of “slop” (assembling pairs with more or less “slop” and preferring the latter). Given the difficult task of pinning down what we mean by “slop” and trying to automatically measure it, we leave implementation of such posttraining informed by slop measurement for future work. We note that we do provide an initial attempt at training a reasoning model in Section 6, and show modest improvements from including our data.
>
> **W2:** We feel there may be a misunderstanding here: We do exactly this (i.e., use existing automatic metrics enumerated in Table 6) to attempt to measure “slop” automatically in Section 6 and observe how the measurement of “slop” interacts with prior work measuring properties of text such as diversity and complexity. In particular, we use these to measure each attribute and then define a linear model (6.1) with these as predictors.
>
> **W3:** We follow best practices and prior methodology for our interview and coding process (e.g., [1, 2]), including an iterative step where all authors independently reviewed the codes and reached agreement. We believe this taxonomy is sufficient in capturing the definitions collected from our experts. Our annotated data is used to train models, where we observe a modest improvement in our automatic evaluations, in line with existing work. This is a first attempt at systematically establishing a taxonomy of slop, and future work should focus on validation with non-experts at scale.
>
> **W4:** “Slop” is a multifaceted construct so we conducted very focused, extensive interviews with our copy editors to ensure alignment on the annotation process. These interviews informed how experts evaluate and edit text for professional settings such as news outlets. We have updated the manuscript **(Appendix E)** with information about our annotators:
> - All native English speakers
> - 30, 20, and 15 years of experience working as Copy Editors
> - Working in education (teaching writing, editing), professional publication (print journalist/writer), and business contexts (Writer, Analyst in print and online)
>
> **W5:** We apologize if this was unclear. For span-level agreement, we closely follow methodology established in prior work [1]. For each annotator, we take the set of spans they selected (character offsets) and compute the proportion of characters that overlap with any span selected by the other annotators.
> ```
> gold_masks  = []
> pred_masks  = []
>
> for each row in df:
>     text = row.text
>
>     gold_spans = parse_spans(row.gold)
>     gold_mask  = mark_characters(text length, gold_spans)
>     gold_masks.append(gold_mask)
>
>     pred_spans = parse_spans(row.pred)
>     pred_mask  = mark_characters(text length, pred_spans)
>     pred_masks.append(pred_mask)
>
> # averages over all rows
> g = concatenate(gold_masks)
> p = concatenate(pred_masks)
>
> tp = count_positions(where g==1 AND p==1)
> fp = count_positions(where g==0 AND p==1)
> fn = count_positions(where g==1 AND p==0)
>
> precision = tp / (tp + fp) if (tp + fp) > 0 else 0
> recall = tp / (tp + fn) if (tp + fn) > 0 else 0
> f1 = 2 * precision * recall / (precision + recall)
>
> return all computed metrics
> ```
>
> We have updated the manuscript to include the pseudocode for the evaluation in **Appendix J**, and will release the code from the evaluations along with the data (and annotations) for reproducibility.
>
> **W6:** We note that this is part of our findings here: To enable large-scale studies, we call for better metrics to evaluate what we have found to be important factors for “slop” assessment. Our hope is that this framework motivates future metric or reward development for the improvement of text quality.
>
> **W7:** Our interpretation is that there is some subjectivity here, but we also emphasize that the axes we have identified do reliably correlate with increased likelihood of texts being perceived as “slop”. Thus while a completely reliable binary assessment of “slop” or “not” may not be feasible (or even possible), our results and analysis suggest that a score quantifying the likelihood that a given piece of text will be comparatively likely to be perceived as “slop” by a reader is feasible; such a score could be used, e.g., as a reward to discourage models from producing “slop”.
>
> (continued)

---

> > ### Author Response · Authors · 2025-11-21
> >
> > (continued)
> >
> >
> > **Q1:** We view this as one of the more interesting observations in this work. In part, these findings provide a secondary validation of our taxonomy; there is a positive correlation between the number of “slop” spans and the final, subjective judgement. We view this finding similar to preference assessments, where individuals may have differing thresholds for a final judgement, and binary labels may be overly reductive. For example, the binary label does not capture any information about the severity of each span found in the text. Our hope is that this provides a foundation for future assessments that more accurately capture the spectrum of low quality text assessments and improve the reliability model evaluations.
> >
> > **Q2:** From our initial definition survey, we had a question asking whether experts believed there to be a difference. From our coding of these responses, we found that the characteristics assigned to sloppy human-writing diverged from our AI-specific codes. For example, features of human-writing focused on incoherence and obscurity of ideas, as well as other aesthetic failures indicating hastily written text (e.g., errors such as typos and poor grammar). We decided to scope our work to examine AI-generated text specifically only as a result. The news dataset had annotations from prior work that contained human-written texts mislabeled as AI-written (people who frequently use ChatGPT for writing tasks are accurate and robust detectors of AI-generated text [3]). We included this subset in our analysis with the assumption that slop spans here would align with codes identified previously for AI-generated text.
> >
> > **Q3:** It’s a great question, but we don’t know! We think the poor performance of GPT-5 (e.g.) here is somewhat surprising, and as the reviewer suggests may be related to biases imposed by RLHF; perhaps because this shapes the model to prefer some of these “slop” characteristics it is for this reason it is unable to reliably discern them? But we can only speculate at this point, and believe exploring this further (based on our findings here) is an interesting direction for future work.
> >
> > [1] Chakrabarty, T., Laban, P., & Wu, C. S. (2025, April). Can ai writing be salvaged? mitigating idiosyncrasies and improving human-ai alignment in the writing process through edits. In Proceedings of the 2025 CHI Conference on Human Factors in Computing Systems (pp. 1-33).
> >
> > [2] Braun, V., & Clarke, V. (2006). Using thematic analysis in psychology. Qualitative Research in Psychology, 3(2), 77–101. https://doi.org/10.1191/1478088706qp063oa
> >
> > [3] Russell, J., Karpinska, M., & Iyyer, M. (2025). People who frequently use ChatGPT for writing tasks are accurate and robust detectors of AI-generated text. arXiv preprint arXiv:2501.15654.

---

> > > ### Author Response · Authors · 2025-11-28
> > >
> > > We wanted to briefly follow up in case you have any questions about our rebuttal. We are happy to offer any further clarification that may be useful. We appreciate your time and engagement!

---

### Official Review · Reviewer_SQw9 · 2025-11-04

**Soundness:** 3
**Presentation:** 3
**Contribution:** 3
**Rating:** 4
**Confidence:** 4

**Summary:**

The paper develops a taxonomy of types of AI "slops" in text from 19 domain experts, conducts span-level annotations on 150 LLM-generated news articles and 100 MS MARCO passages with professionals. The paper also analyzes agreement and identifies what features of the taxonomy are the strong predictors of the AI slops. Lastly, the paper conducts testing of whether machine learning models, LLM-as-judges and rewarding models can identify the AI slops.

**Strengths:**

- Highly timely research problem in LLM generation field. The paper's scope that covers both quantiative and qualitative analyses (e.g. taxonomy development, span-level annotations by domain experts, and analytics) provide several insights of the linguistic and stylistic factors in AI-generated textual contents.

- Glad to see honest reports on negative results in Section 5. The paper carefully shows that standard metrics, LLM judges, and even finetuned models fall short, thus avoiding overclaiming of their findings.

**Weaknesses:**

- The taxonomy development of AI slops is primarily based on two English domains and information-focused texts (News articles and web search queries); the axes of the taxonomy may not be applied to other languages (e.g., Spanish, French, or Chinese) or genres (e.g., creative writing). Also, the demographics of experts who participated in the survey and taxonomy development processes are not clearly mentioned (ie, is English your first language?). Please report the experts' language backgrounds (e.g., whether English is their first language) so reviewers can judge how general the taxonomy axes are.

- Concern about annotation reliability. Multi-label span annotation is a highly complicated task, and several axes show low agreement (some < 0.3 in Appendix Table 12), indicating notable disagreement or sparse overlap among annotators.

**Questions:**

- For LLM judges, did the authors analyze how o4-mini's rationale of span-level annotations line up with the expert-labeled spans and codes? If you have some relevant data, a small qualitative study would help: (1) overlap between LLM rationale spans and human-annotated spans (using token F1) or (2) axe alignment (which "slop" categories the LLM cites most vs humans), etc.

- In Section 5, the authors mentioned that the domain of datasets also affects the evaluation of LLM-generated texts. However, within the News set, I think the magazine/source type and article topic (e.g. original titles) can also shape the slop style in the LLM-generated texts. Have you checked (1) the per-outlet(source) prevalence of each "slop" axis, (2) topic-level effects (science vs politics vs lifestyle) etc?

---

> ### Author Response · Authors · 2025-11-21
>
> We thank the reviewer for their thorough feedback and suggestions, and appreciate that they found this work timely and careful in highlighting limitations of current evaluation methods! We address the weaknesses and answer the questions below. We have also added all suggestions and experiments to the manuscript, written in pink.
>
> **W1:** We thank the reviewer for highlighting this. We surveyed experts from North American institutions (academic and industrial backgrounds); see Appendix Figure 3 for details. We did not explicitly collect native language information but all experts were highly educated and experienced working in English (see Figure 3a) so highly fluent. We collected annotations on English texts only, and acknowledge that this is an inherent limitation of the work. It is possible that the taxonomy wouldl differ in other languages, and while we think it is likely that the hallmarks of “slop” would be consistent, this would need to be verified empirically in future work. We have added our focus on English texts as an explicit limitation in the revised manuscript **(Discussion, L492; Appendix E)**. Our hope here is to provide an initial foundation to enable such follow-up analyses.
>
> **W2:** We agree that span annotation tasks are complicated, especially for a nebulous (but important) concept like “slop” and the factors that correlate with it. We interpret low agreement as an informative signal about conceptual ambiguity: this informed our decision to iteratively update the taxonomy to collapse codes that alone showed low agreement, but higher stability under a higher-order concept (i.e., Table 6).
>
>
> **Q1: Re: Rationale Overlap**
>
> This is an important observation and we agree that a qualitative assessment of the generated rationales and the alignment of the selected codes would strengthen our analysis. We note that our annotators do not provide a rationale for each highlighted span, but instead provide an editor’s note at the end of each annotation task, an industry standard comprising a top-level assessment of the text (which will be included in the released dataset to facilitate future evaluations). Since we cannot compare these notes directly to the individual span rationales, we look at the generated reasoning chains (i.e., (2) Axe alignment) of the few-shot setting (k=0 and k=5), and visualize trigram counts against the human-label distribution.
>
> **Section 6.3.1 LLM-GENERATED SPAN RATIONALES:** Here, we discuss the findings of the axis alignment. We observe that the o3-mini rationales overly represent issues with Information Density (n=80), while missing other axes identified frequently by humans (Coherence, Tone, Bias, Relevance, and Factuality). A qualitative assessment of the tri-grams **(Appendix Figure 8)** shows issues of information density and templatedness appearing most frequently. It is interesting that the human-annotated distribution of Density is larger relative to the other codes (n=25), indicating an alignment on the importance or prevalence of this code, but we believe this provides a strong foundation for guiding future improvements for alignment of judge-based metrics.
>
> We also notice that, while increasing the shots guides the reasoning to qualitatively align better with the human-written responses, the actual spans selected remain fairly similar (word-level overlap) even when providing examples in context:
>
> k0 vs k5 word-level overlap:
> precision: 0.65
> recall:    0.61
> f1: 0.63
>
> We have added this as an additional analysis in the manuscript.
>
>
> **Q2: Re: Per-outlet prevalence, topic-level effects**
>
> We agree that there are interesting, finer-grained factors within the domain that may influence the style of text; however, similarities in text purpose would still be captured with the coarser discussion of domain. Given the scale of the experiments, we cannot reliably conclude any effects from the topic within the domain, but we do plot the prevalence of the axes split by source and note the counts of articles within each source **(Appendix Figure 9, Table 13)**. We note interesting per-source features, for example, articles from The Smithsonian (n=38) have fewer issues with Factuality and Bias, compared to articles generated for the New York Times (n=37).
>
> Importantly, we note that the overall “slop” categories are roughly represented similarly across sources: Style Quality codes appear at a much higher rate relative to Information Quality within each. Information Utility is also annotated relatively higher across sources compared to Information Utility.  We now include this discussion in **Appendix G**, and note this now on **L353**. All information (including Source and Topic) will be provided in the code and data release.

---

> > ### Comment · Reviewer_SQw9 · 2025-11-26
> >
> > Thank you for the additional analyses and clarifications that you've made within this short timeframe. I have adjusted my scores. Please reflect your additional analyses on the current manuscript.

---

> > > ### Author Response · Authors · 2025-11-28
> > >
> > > Thank you for taking the time to read our rebuttal and providing further feedback! We appreciate your engagement and have updated our current manuscript to include the additional analysis.

---

### Author Response · Authors · 2025-11-21

Overall, we are pleased that the reviewers agree that this work addresses a timely and relevant problem in LLM evaluations (SQw9, zmra, XZAE). We are especially encouraged that the reviewers find this work provides a structured language for text evaluations beyond vague, binary assessment (XZAE), a foundational framework for future work (8fEQ), and is an appropriate, careful quantitative study over stylistic and linguistic factors (SQw9, zmra).

We are grateful to the reviewers for their valuable comments and suggestions, and would like to address the common comments here. **We note that we have temporarily added changes to the manuscript in pink to highlight additional experiments and modifications following reviewer suggestions.**

**1. Annotation Reliability; Background (SQw9, XZAE, 8fEQ, zmra)**

We follow best practices and prior work (e.g., [1, 2] to carefully conduct our coding and annotation assessments. We selected annotators with 15-30 years of copy-editing experience, and conducted multiple training and adjudication sessions to align editing styles. We now report details on annotator backgrounds in Appendix E. We also take care to report several agreement metrics over each facet of our annotation (binary label, spans, codes; L241, 247, 264) that provide a realistic assessment in paradigms that have few-rater designs [3]. We find sufficient and consistent labels to enable a systematic first pass at defining the “slop” construct. We release our data and code to encourage larger-scale validation with non-experts in future work.


**2. English-only Evaluations (SQw9, 8fEQ)**

We collect annotations on only English texts and acknowledge this is an inherent limitation of the work, given the difficulty of verifying annotation quality due to our background as authors. We expect the hallmarks of “slop” to be consistent across languages, but would need to be empirically verified in future work. We believe our work provides data and a framework to facilitate comparisons of “slop” evaluations in other languages. We now explicitly acknowledge this in our limitations section (L537).


[1] Braun, V., & Clarke, V. (2006). Using thematic analysis in psychology. Qualitative Research in Psychology, 3(2), 77–101. https://doi.org/10.1191/1478088706qp063oa

[2] Artstein, R., & Poesio, M. (2008). Inter-coder agreement for computational linguistics. Computational linguistics, 34(4), 555-596.

[3] Marchal, M., Scholman, M., Yung, F., & Demberg, V. (2022, October). Establishing annotation quality in multi-label annotations. In Proceedings of the 29th international conference on computational linguistics (pp. 3659-3668).

---

### Author Response · Authors · 2025-11-30
**Discussion Summary for AC**

We would like to summarize the discussion prior to the changes. In our rebuttal, we responded to all reviewer comments and clarified our positioning of the work as an empirically-validated foundation based on expert analysis. We also note that we will release the annotated data and code as part of the contribution (see Supp. Materials).

One reviewer engaged with the discussion and increased their score after reading our clarifications. The other three reviewers were unable to follow up before the discussion period ended.

**(1) RE: Language, Expert Demographics:** We explicitly address English as an inherent limitation of the work, and confirm our annotators are highly fluent Native English speakers. We also added detailed annotator backgrounds highlighting their 15 to 30 years of experience as professional writers and copy editors.

**(2) RE: Annotation Reliability:** We stress that span annotation of a multi-faceted construct is conceptually ambiguous, but our agreement scores are nonetheless informative. We carefully report many agreement metrics appropriate for few-rater designs. This motivates methodology to combine codes and highlight areas for future work.

**(3) RE: Additional Analyses of LLM Responses:** We add additional analyses of o3-mini rationales across spans and axes showing that LLMs overly represent Density issues. This highlights a need for careful usage of LLMs as judges in this domain. We now include the full analysis as an additional section in our manuscript.

**(4) RE: LLMs as Slop Evaluators:** We highlight that our existing metrics, LLMs as judges and traditional text evaluations, fall short in reliably predicting all "slop" constructs. We believe this is an important finding and emphasizes the importance of future work developing more reliable measurements that accurately capture text quality.

**(5) RE: Interpretation of Limited Annotator Pool:** We acknowledge the reviewers' concerns but emphasize that our annotators are highly trained experts, and that we followed best practices and prior work for this study. For example, we conducted multiple calibration and adjudication sessions, and we report multiple agreement metrics that are more robust to few-rater designs. We stress that our data release is intended precisely to support future studies and scaling beyond expert evaluations.

**(6) RE: Reframing the Contribution:** We have slightly revised the abstract and introduction to now better reflect the paper's contributions: first identifying the gap between human slop judgements and existing metrics, second establishing a taxonomy grounded in expert definitions, and third providing initial empirical attempts. We are careful to not claim a fully solved and scalable method, but rather focus on providing the foundation and data for future progress.

We hope this context is useful in the final evaluation.

---

### Meta-Review · Area_Chair_Rdut · 2026-01-07

**Summary:**

- Reviewers SQw9 and 8fEQ were concerned that being limited to only English and two domains (news and QA) may limit the quality of the taxonomy.
- Reviewers SQw9 and zmra ask for more background on the annotators.
- Reviewers SQw9, zmra, and XZAE were concerned about limited inter-annotator agreement and annotation best practices.
- Reviewers zmra, 8fEQ and XZAE raised concerns about the inability of automatic slop detection, which might imply that the proposed framework is insufficient to robustly measure slop.

**Reviewer Concerns:**

- The authors explicitly note both English-only and additional domains as a limitation and possible future work.
- The authors describe the annotators' backgrounds and the best practices they follow.
- The author reframe the contribution as a foundational step for future progress.

I believe that the fundamental idea of defining slop was successful, but the newly limited scope of just an initial step is insufficient for an ICLR publication. I recommend submission to NAACL or ACL.

**Reviewer Scores:**

- SQw9: would have increased score (+2).
- zmra: would have increased score (+1)
- 8fEQ: would not have increased score.
- XZAE: would not have increased score.

---

### Decision · Program_Chairs · 2026-01-26

Reject